

# On the construction of charged operators
# inside an eternal black hole

Monica Guica[1,2] and Daniel Jafferis[3]

**1** Nordita, KTH Royal Institute of Technology and Stockholm University,
Roslagstullsbacken 23, SE-106 91 Stockholm, Sweden
**2** Department of Physics and Astronomy, Uppsala University,
SE-751 08 Uppsala, Sweden
**3** Center for the Fundamental Laws of Nature, Harvard University,
17 Oxford St, Cambridge MA, 02138, USA

## Abstract

We revisit the holographic construction of (approximately) local bulk operators inside an eternal AdS black hole in terms of operators in the boundary CFTs. If the bulk operator carries charge, the construction must involve a qualitatively new object: a Wilson line that stretches between the two boundaries of the eternal black hole. This operator - more precisely, its zero mode - cannot be expressed in terms of the boundary currents and only exists in entangled states dual to two-sided geometries, which suggests that it is a state-dependent operator. We determine the action of the Wilson line on the relevant subspaces of the total Hilbert space, and show that it behaves as a local operator from the point of view of either CFT. For the case of three bulk dimensions, we give explicit expressions for the charged bulk field and the Wilson line. Furthermore, we show that when acting on the thermofield double state, the Wilson line may be extracted from a limit of certain standard CFT operator expressions. We also comment on the relationship between the Wilson line and previously discussed mirror operators in the eternal black hole.



# 1  Introduction and summary

One of the most remarkable aspects of the AdS/CFT correspondence is that it gives us a definition of quantum gravity in anti-de Sitter space-time [1]. However, while the holographic dictionary for extracting CFT quantities as boundary limits of bulk ones is relatively straightforward, it is far more challenging to reconstruct the physics of the AdS interior from the CFT. In certain cases - such as vacuum AdS - there is a perturbative procedure [2–11] to determine bulk operators from highly nonlocal boundary ones, which may be possible to resum non-perturbatively to well-defined CFT operators. However, if the bulk region lies behind the horizon of an AdS black hole, [12–14] have argued that the CFT description of interior bulk operators is *state-dependent*, which means that the CFT operator that represents the bulk field can depend sensitively on unmeasured details of the quantum microstate of the black hole.

State-dependent operators are invoked when there does not exist a fixed CFT operator that has the properties inferred from bulk perturbation theory (e.g., behaving as a local operator, obeying a particular algebra) in all the states in which such a behaviour is expected [14]. A "state-dependent" CFT operator associated to a particular black hole microstate state $|\Psi\rangle$ is then only required to act "nicely" in a small subspace - denoted $\mathcal{H}_\Psi$ - of the total CFT Hilbert space, which consists of $|\Psi\rangle$ and not-too-large excitations thereof; by construction, $\mathcal{H}_\Psi$ corresponds precisely to the part of the CFT Hilbert space that can be probed by an observer in the bulk.

While state-dependence is a very interesting proposal for a concrete implementation of black hole complementarity, it takes as an input the bulk perturbative description, including smoothness of the horizon. This led [15] to consider the issue of state-dependence in the eternal black hole, dual to to the thermofield double state of two CFTs [25], which is believed to have a smooth horizon. By considering a set of time-shifted states that correspond to the same background geometry, [15] were able to exhibit state-dependence also in this case.

In this work, we revisit the holographic dictionary in the eternal black hole background, with the aim of better understanding the mechanism responsible for state dependence. Rather than studying gravitational interactions in the bulk, we concentrate on the simpler case of charged scalars coupled to bulk electromagnetism. By carefully taking into account issues related to gauge invariance and boundary conditions, we uncover a new element of the holographic dictionary: a boundary-to-boundary Wilson line, and discuss its relation to state-

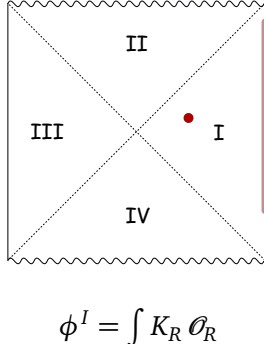

$$\phi^I = \int K_R \, \mathcal{O}_R$$

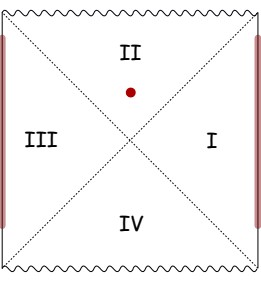

$$\phi^{II} = \int K_L \, \mathcal{O}_L + \int K_R \, \mathcal{O}_R$$

Figure 1: Naïve representation of a charged scalar in regions I, II of the eternal black hole in terms of smeared CFT operators on the two boundaries.

dependence. This object has been previously considered in [16] as a quantitative probe of the ER=EPR conjecture [17]. In the following, we give a brief account of how the Wilson line operator appears, and of its expected properties.

We set out to understand the representation of a charged (scalar) bulk operator[1] $\phi(y)$ placed inside an eternal black hole in terms of CFT operators on the two boundaries. The dual operator to this bulk field in the left/right CFT is denoted as $\mathcal{O}_{L/R}$ and carries charge $q$ under the left/right conserved $U(1)$ charges, $Q_{L/R}$. Since all points in the eternal black hole are in causal contact with at least one of the two boundaries, it would seem that all light bulk fields can be obtained by smearing the local CFT operators $\mathcal{O}_{L/R}$ on the two sides, as pictured in figure 1; there, $\phi(y)$ represents the charged bulk scalar, $K_{L/R}(y|x_{L/R})$ are bulk-to-boundary propagators from the bulk point $y$ to the boundary point $x_{L/R}$, and the integrals run over the respective boundaries.

However, it is easy to see that these naïve expressions violate charge conservation as we move the bulk field from region I to region II of the black hole, since the expression $\phi^I$ for the bulk field in region I has zero commutator with the charge $Q_L$ in the left CFT, whereas the expression $\phi^{II}$ for the bulk field in the interior has a non-zero commutator (see also [14]). The problem is easy to identify: we need to consider a gauge-invariant bulk operator[2], as the CFT only captures gauge-invariant data in the bulk. The gauge-invariant bulk operator that we will study throughout this paper is a charged scalar field $\phi(y)$, connected via a Wilson line to a point $\widehat{x}_R$ the right boundary[3]

$$\hat{\phi}(y) = \phi(y) P \exp(iq \int_\Gamma A), \tag{1}$$

where $\Gamma$ is a bulk path that starts at $y$ and ends at $\widehat{x}_R$. This is shown in figure 2. Note that, due to the framing, this operator is not exactly local in the bulk.

The commutation relations of $\hat{\phi}$ with the boundary charges $Q_{L/R}$ are entirely determined by the boundary endpoint of the Wilson line; in our setup, $\hat{\phi}(y)$ has $Q_L = 0$ and $Q_R = q$, irrespective of where the bulk point $y$ is located. From the bulk point of view, the charges work out correctly because the gauge field appearing in (1) contributes at leading order to

---

[1]Our notation is as follows: $y^M$ are bulk coordinates, with $M = 1,\dots,D = d+1$, $x^\mu = (t, x^i)$ are boundary coordinates and $z$ denotes the radial direction. Coordinates on the left/right boundaries are denoted by $x^\mu_{L,R}$.

[2]One may argue that $\phi(y)$ does correspond to a gauge-invariant bulk operator if we work in radial gauge, since then $\phi(y) = \hat{\phi}(y)$ for a Wilson line that stretches along the radial direction. However, as we will explain, radial gauge is disallowed in the eternal black hole background, which is why we consider $\hat{\phi}$ (see also [18]).

[3]Other framings are also possible (including smeared ones as e.g. the one corresponding to the charged operator in Coulomb gauge), but we will not consider them here.

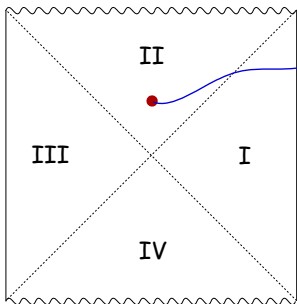

Figure 2: The charged scalar connected to the right boundary via a Wilson line is a gauge-invariant bulk operator, carrying charges $Q_L = 0$ and $Q_R = q$.

the Dirac bracket commutator of the charges[4] with $\hat{\phi}$. It thus becomes intuitively clear that in order for the boundary representation of the field in region II to have the correct charge, we should multiply the contribution of the left operators in figure 1-right to $\hat{\phi}$ by a boundary-to-boundary Wilson line

$$W_{LR}(\widehat{x}_L | \widehat{x}_R) = \mathscr{P} \exp\left( iq \int_{\widehat{x}_L}^{\widehat{x}_R} A \right). \tag{2}$$

This object has charge $-q$ on the left and $+q$ on the right, and thus $\mathscr{O}_L W_{LR}$ has the correct charges. The boundary representation of the bulk operator will schematically take the form

$$\hat{\phi}(y) = \int d^d x_R K_R(y | x_R) \mathscr{O}_R^{(j)}(x_R) + W_{LR}(\widehat{x}_L | \widehat{x}_R) \int d^d x_L K_L(y | x_L) \mathscr{O}_L^{(j)}(x_L). \tag{3}$$

In the above expression, $\mathscr{O}_{L/R}^{(j)}$ denote the charged operators on the left/right boundaries, dressed by arbitrary powers of the respective current. Such expressions were shown in [7] to appear in the boundary representation of a charged bulk field and, as we review in section 2, all powers of $j$ contribute at the same order to the commutator of $\hat{\phi}$ with the boundary currents. The particular dressing by the currents in $\mathscr{O}_{R,L}^{(j)}$ depends on the shape of the bulk Wilson line and on its endpoints $\widehat{x}_{L,R}$. The point $\widehat{x}_L$ is an arbitrarily chosen common point for all the Wilson lines that frame the left operators $\mathscr{O}_L(x_L)$ to the point $\widehat{x}_R$ on the right boundary; such a common point can always be chosen by appropriately adjusting the current dressing of $\mathscr{O}_L^{(j)}$.

Thus, in presence of two boundaries, the expression for a charged operator inside the black hole must contain a contribution from a new gauge-invariant operator: a boundary-to-boundary Wilson line $W_{LR}$, in addition to the well-known boundary operator contributions, dressed and smeared. Its existence can be easily shown via a careful analysis of the equations of motion on a manifold with two boundaries, when contributions from the gauge field are included.

We would now like to find the representation of this new object in terms of operators in the boundary CFTs. Despite being a purely gauge field configuration, the Wilson line cannot be constructed just from the boundary currents, because the latter do not carry electric charge. To better understand what happens, suppose for simplicity that all the CFT currents have been turned off, so we have an everywhere flat gauge field, $A = d\lambda$. In a two-sided geometry, the boundary-to-boundary Wilson line is given by

$$\langle W_{LR} \rangle = e^{iq \int_L^R A} = e^{iq(\lambda_R - \lambda_L)}, \tag{4}$$

---

[4]It is not hard to see that all operators linear in $\phi$ but containing arbitrary powers of the gauge field contribute at the same order in the (small) coupling constant to the commutator with the boundary currents.

where $\lambda_{L,R}$ are the values of the gauge parameter on the two boundaries[5]. While the individual values of $\lambda_{L,R}$ are not meaningful because they can be changed by a constant overall gauge transformation, their difference *is* gauge invariant and corresponds to a new mode of the gauge field that only exists in two-sided geometries. Following the usual AdS/CFT logic, this new gauge-invariant mode should be associated with some CFT operator. This operator, which we denote by[6] $\varphi$, does have non-trivial commutators with the boundary charges, as can be deduced from the transformation properties of $\lambda_R - \lambda_L$ under boundary global gauge transformations

$$[Q_L, \varphi] = -i\,, \quad [Q_R, \varphi] = i\,. \tag{5}$$

More generally, $\varphi$ is defined as

$$\varphi(\widehat{x}_L, \widehat{x}_R) = \int_\Gamma A\,, \tag{6}$$

where the curve $\Gamma$ stretches between a point $\widehat{x}_L$ on the left boundary and a point $\widehat{x}_R$ on the right boundary. Since the gauge group is compact, we have $\varphi \sim \varphi + 2\pi$, and thus this operator does not make sense in the full Hilbert space; however, its action is well defined in a small neighbourhood of the state of interest. The full Wilson line is $W_{LR} = e^{iq\varphi}$, regulated by appropriate counterterms.

The operator $\varphi$ will be very useful in our discussion, as it is much simpler to study than the exponentiated Wilson line. First, its derivatives are linear in the CFT currents, which means that all but its zero mode (discussed above) can be reconstructed from them. Second, for an appropriate choice of the curve $\Gamma$, as in the example of section 3.3, $\varphi(\widehat{x}_L, \widehat{x}_R)$ behaves as a local operator from the point of view of either boundary, by which we mean that its commutator with local operators in the left/right CFT vanishes outside the lightcone associated with $\widehat{x}_{L/R}$. Finally, for $D = 3$ and at low energies, $\varphi$ behaves as a non-chiral free boson whose left/right-moving pieces come from the left/right boundary, with a shared zero mode. This is the same as the behaviour of pure three-dimensional Chern-Simons theory on a manifold with two boundaries [22]. All these properties of $\varphi$ are inferred from bulk perturbation theory in a two-sided black hole geometry.

Next, we would like to argue that there is no fixed CFT operator acting as $\varphi$ (or its exponentiated version) on the product Hilbert space of the two CFTs. This would imply that the Wilson line is a state-dependent operator, allowing us to make a connection with the statements of [15]. This seems to be intuitively clear from the fact that $\varphi$ (and in particular, its zero mode) is only defined in entangled states dual to connected two-sided geometries. Since the set of such entangled states is a non-linear subspace of the total Hilbert space, the Wilson line cannot be represented by a linear operator. We can in fact prove state-dependence, along the lines of [15], by studying the action of the Wilson line on arbitrary time-shifted states.

In order to make this argument, however, we first need to determine the action of the Wilson line on the small Hilbert space built around the state of interest - in our case, the thermofield double state. We present two methods to do so.

The first method is to simply find the action of the Wilson line on every element of $\mathscr{H}_{\Psi_{\text{tfd}}}$, which abstractly defines it as an operator; this is in the same spirit as the usual definition of mirror operators [13]. The action of the Wilson line on $\mathscr{H}_{\Psi_{\text{tfd}}}$ can be entirely determined from its commutators (around $\mathscr{H}_{\Psi_{\text{tfd}}}$) with the low-lying CFT operators and its action on the thermofield double state. The former can be inferred from bulk perturbation theory, whereas the latter can be obtained from a path integral argument.

The second method is inspired from the fact that the total Hilbert space of the system is the tensor product of the left and the right CFT Hilbert spaces. Thus, an operator of definite

---

charges $Q_L = -q$, $Q_R = +q$ should be decomposable as a sum of products of a charged operator from the left, and a charged operator from the right. Around the thermofield double state, there is a pictorial way to realize this decomposition of the Wilson line by representing it as the fusion, at the bifurcation surface of the eternal black hole, of a negatively charged operator framed to the left boundary with a positively charged operator framed to the right. As the two bulk insertion points approach each other, a divergence develops, and the Wilson line can be extracted from the coefficient of this divergence. Note that in general entangled states (e.g., dual to geometries without a bifurcation surface) no such divergence is expected for operators inserted near the intersection of the future and past horizons on each side, showing that this construction is extremely sensitive to the state of the system.

[5]In $D > 3$, these values have to be constants due to the boundary conditions on the gauge field. In $D = 3$, they must be constants in order to have a zero expectation value for the currents. See sections 2, 3.

[6]Strictly speaking, this will be just the zero mode of $\varphi$. The full definition of $\varphi$ is (6).

The plan of this paper is as follows. In sections 2, 3 we work out the expression for the gauge-invariant bulk field $\hat{\phi}$ in terms of CFT operators in several concrete examples and show the appearance of the Wilson line. We use bulk perturbation theory to infer some properties of the dual operator. In section 4, we discuss the CFT representation of the Wilson line when acting on the thermofield double state, first - by computing its action on the thermofield double state, and then - by constructing it via OPE fusion at the bifurcation point. We also discuss the relation between the Wilson line and the results of [15].

As this work was nearing completion, [19] appeared, which has some overlapping statements.

## 2 Charged scalar coupled to $D = 3$ Chern-Simons

In this section, we consider the simplest possible example - that of a $U(1)$ Chern-Simons gauge field in three dimensions coupled to charged scalar field $\phi$. The action is

$$S = \int d^3x \sqrt{g} \left( \frac{k}{8\pi} \epsilon^{\mu\nu\rho} A_\mu \partial_\nu A_\rho - D_\mu \phi\, D^\mu \phi^\star - m^2 |\phi|^2 \right), \tag{7}$$

where $D_\mu = \partial_\mu - iqA_\mu$.

We are interested in the boundary representation of the gauge-invariant bulk scalar $\hat{\phi}$ defined in (1). We first show (in section 2.1) that the bulk equations of motion, upon perturbatively including the contributions from the bulk gauge field, lead to an expression of precisely the form (3) for $\hat{\phi}$. This expression contains a contribution from the boundary-to-boundary Wilson line. In section 2.2 we discuss the holographic interpretation of the Wilson line. Finally, in 2.3, after carefully discussing the choice of gauge, we work out the Dirac brackets of the Wilson line with the bulk gauge field and scalar operators. Upon quantization, these will yield the commutators of our newly-found operator with the usual low-lying CFT operators around states dual to smooth two-sided geometries.

### 2.1 Analysis of the wave equation

To obtain the representation of the gauge-invariant bulk scalar $\hat{\phi}$ in terms of CFT operators, one needs to perturbatively solve the equations of motion for $\phi$ and the gauge field. The equations of motion derived from (7) read

$$(\Box - m^2)\phi = iq(\phi \nabla_\mu A^\mu + 2A^\mu \partial_\mu \phi) + q^2 A^2 \phi\,, \qquad F_{\mu\nu} = -\frac{4\pi}{k} \epsilon_{\mu\nu\lambda} J^\lambda\,, \tag{8}$$

where the conserved current is given by

$$J_\mu = iq(\phi^\star D_\mu \phi - \phi (D_\mu \phi)^\star)\,. \tag{9}$$

Note that the right-hand-sides (RHS) of the above equations are quadratic or higher in the basic fields. At zeroth order, we can just neglect the RHS and the solution is

$$\phi^{(0)}(y) = \int d^2x' K^{(\phi)}(y|x')\mathcal{O}(x')\,, \qquad A_\mu^{(0)}(y) = \frac{2}{k}\int d^2x' K^{(A)}(y|x')j_\mu(x')\,, \qquad (10)$$

where $K^{(\phi,A)}(y|x)$ are appropriate bulk-to-boundary propagators for the scalar and the gauge field, respectively[7]. The higher order contributions are obtained by including the interaction terms on the RHS of (8); for example, the term linear in $q$ leads to a correction [10]

$$\phi^{(1)}(y) = iq \int d^3y' \sqrt{g(y')} G^{(\phi)}(y|y')[\phi^{(0)}(y')\nabla_\mu A_{(0)}^\mu(y') + 2A_{(0)}^\mu(y')\partial_\mu\phi^{(0)}(y')]\,, \quad (11)$$

where $G^{(\phi)}(y|y')$ is the bulk-to-bulk Green's function for $\phi$. Plugging in the expressions (10) for the zeroth order fields, we find (11) corresponds to a set of multitrace boundary operators of the schematic form [8]

$$\frac{1}{k} : \partial^{\mu_1 \dots \mu_p} \Box^m j^\nu \partial_{\mu_1 \dots \mu_p} \Box^n \partial_\nu \mathcal{O} : \qquad (12)$$

The full expression for $\hat{\phi}$ is obtained by summing the perturbative (in $q$ and $1/k$) contributions from the bulk scalar and the Wilson line piece.

It is common, when discussing the construction of bulk operators from the CFT perspective, to discard all multitrace operators coming from the interaction terms in the Lagrangian, on the basis that when $k$ is large, their contribution to correlation functions is negligible. However, it is not hard to see that this is no longer true if one considers the OPE of the bulk field with the CFT current. Indeed, from the OPEs

$$j(z)j(0) \sim \frac{k}{2z^2}\,, \qquad j(z)\mathcal{O}(0) \sim \frac{q}{z}\mathcal{O}(0) \qquad (13)$$

it is clear that the OPE of $j$ with $\phi^{(1)}$ scales in the same way as that of $j$ with $\phi^{(0)}$. The lesson we draw from this analysis is that, if we want to have a boundary representation of the bulk scalar that correctly takes into account the charge of the operator, we cannot just discard the interaction terms on the RHS of (8).

However, it is not hard to see that the interaction terms on the RHS of the gauge field equation in (8) are strictly subleading in the large $k$ limit, and thus *can* be consistently discarded. This corresponds to taking the $k \to \infty$ limit with $q$ kept fixed. In this case, we can take the gauge field to solve

$$F_{\mu\nu} = 0 \quad \Rightarrow \quad A_\mu = A_\mu^{(0)} = \partial_\mu\lambda\,. \qquad (14)$$

Then, neglecting the gravitational backreaction ($N \to \infty$) and all possible (self-)interactions of the scalar, the solution for the gauge field continues to be pure gauge, whereas the solution for $\phi$ can be obtained perturbatively from (8)

$$\phi = \phi^{(0)} + \phi^{(1)} + \phi^{(2)} + \dots\,, \qquad (15)$$

where

$$(\Box - m^2)\phi^{(n)} = iq\left(\phi^{(n-1)}\nabla_\mu A_{(0)}^\mu + 2A_{(0)}^\mu\partial_\mu\phi^{(n-1)}\right) + q^2 A_{(0)}^2\phi^{(n-2)}\,. \qquad (16)$$

Note that the resulting boundary expression will be linear in $\mathcal{O}$, but will contain all possible powers of the current. To find the expression for the gauge-invariant scalar operator $\hat{\phi}$, one

---

[7]To define $K^{(A)}$, one first needs to fix gauge that completely determines $A$ in terms of the boundary data. In the two-sided black hole, these propagators have contributions from both boundaries.

additionally needs to include, perturbatively, the contributions of the bulk-to-boundary Wilson line in (1).

Applying the above procedure, one finds that the boundary representation of $\hat{\phi}$ defined in (1) at linear order in $\mathcal{O}$ and *all* orders in the current is given by

$$\hat{\phi}(y) = \int d^2x' \, K^{(\phi)}(y|x') \, e^{iq[\lambda(\hat{x}) - \lambda(x')]} \, \mathcal{O}(x') \,, \tag{17}$$

where $\lambda$ has been defined in (14). This expression matches the gauge transformation of $\hat{\phi}$, as the bulk field is represented by boundary operators that only transform under gauge transformations at $\hat{x}$. A simpler way to derive the above expression would be to note that in the $k \to \infty$, $q$ fixed limit, $\hat{\phi}$ satisfies the free wave equation

$$(\Box - m^2)\hat{\phi} = 0 \,, \tag{18}$$

obtained by plugging in $A = A^{(0)} = d\lambda$ into the equation of motion (8). Thus, $\hat{\phi}$ can be written as the usual smeared expression of boundary operators of the form

$$\hat{\mathcal{O}}(x|x_0) \equiv \mathcal{O}(x) \, e^{iq[\lambda(\hat{x}) - \lambda(x)]} \,. \tag{19}$$

Suppose now we have two boundaries, and that the Wilson line $\Gamma$ is connected to some point $\hat{x}_R$ on the right boundary. If the bulk operator is inside the horizon, then the smearing function $K$ has support on both boundaries, and we have

$$\hat{\phi}(y) = \int d^2x_L \, K_L(y|x_L) \, e^{iq[\lambda_R(\hat{x}_R) - \lambda_L(x_L)]} \, \mathcal{O}_L(x_L) + \int d^2x_R \, K_R(y|x_R) \, e^{iq(\lambda_R(\hat{x}_R) - \lambda_R(x_R))} \, \mathcal{O}_R(x_R)$$

$$= W_{LR}(\hat{x}_L, \hat{x}_R) \int d^2x_L \, K_L(y|x_L) \, \mathcal{O}_L^{(j)}(x_L, \hat{x}_L) + \int d^2x_R \, K_R(y|x_R) \, \mathcal{O}_R^{(j)}(x_R, \hat{x}_R) \,, \tag{20}$$

where $\lambda_{L/R}$ are the values of the gauge parameter at the left/right boundary and

$$W_{LR}(\hat{x}_L, \hat{x}_R) = e^{iq[\lambda_R(\hat{x}_R) - \lambda_L(\hat{x}_L)]} \,. \tag{21}$$

This expression precisely coincides with (3) and shows explicitly the way in which the boundary-to-boundary Wilson line is entering the computation. For simplicity, we have chosen the left operators to be all connected to some arbitrarily chosen point $\hat{x}_L$ on the left boundary. The dressed operators on the left/right boundaries are, in this case

$$\mathcal{O}_{L/R}^{(j)}(x_{L/R}, \hat{x}_{L/R}) = e^{iq(\lambda_{L/R}(\hat{x}_{L/R}) - \lambda_{L/R}(x_{L/R}))} \mathcal{O}_{L/R} = e^{iq \int A_{L/R}^{\partial}} \mathcal{O}_{L/R} \,, \tag{22}$$

where in the last term we have rewritten the argument of the exponential as an integral over the gauge field on the boundary, running from $x_{L/R}$ to $\hat{x}_{L/R}$. Thus, in three dimensions with $k \to \infty$, the dressing of the charged boundary operators $\mathcal{O}$ by the currents is very simple - just a Wilson line running along the respective boundary. This is represented in figure 3.

Note that since the bulk gauge field in three-dimensional Chern-Simons theory is pure gauge in our approximation, the Wilson line only depends on the value of the gauge parameter at the boundaries, and not on the shape of the Wilson line in the bulk.

## 2.2 Holographic interpretation

In the above discussion, $\lambda$ is the classical gauge parameter, subject to appropriate boundary conditions. Of course, in order to obtain the CFT representation of $\hat{\phi}$, we need to trade $\lambda$ for the appropriate boundary operators, using the holographic dictionary.

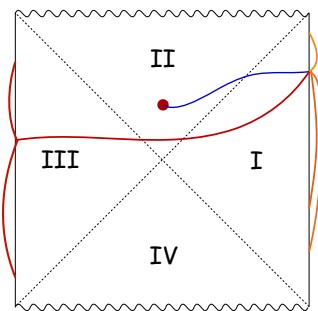

Figure 3: Expression for the gauge-invariant bulk scalar $\hat{\phi}$ (blue line) in terms of the smeared dressed right operators (orange lines) and right-framed left operators (red lines). The left contributions can be decomposed into a dressed operator contribution and a boundary-to-boundary Wilson line.

The bulk Chern-Simons field $A = d\lambda$ is holographically dual to a holomorphic, conserved two-dimensional CFT current. Consequently, it is natural to use light-cone coordinates on the boundary, $x^{\pm} = (x \pm t)/\sqrt{2}$, when working in Lorentzian signature. The radial bulk coordinate will be denoted by $z$, with boundary(-ies) located at $z = z_{\alpha}$.

Remember that in pure three-dimensional Chern-Simons theory, $A_+$ and $A_-$ are canonically conjugate to each other, and thus only one of them can fluctuate. Setting $A_- = \partial_- \lambda = 0$, we have [20]

$$\langle j_+^{(\alpha)}(x^+) \rangle = \frac{k}{2} A_+(x^+, z_\alpha) \, , \tag{23}$$

where $j^{(\alpha)}$ is the CFT current on the boundary at $z_\alpha$. Since $A_+(x^+, z_\alpha) = \partial_+ \lambda(x^+, z_\alpha)$, $\lambda(x^+, z_\alpha)$ should correspond to a putative "chiral boson" operator $\widetilde{\varphi}_\alpha(x^+)$, which by definition satisfies

$$\frac{k}{2} \partial_+ \widetilde{\varphi}_\alpha(x^+) = j_+^{(\alpha)}(x^+) \tag{24}$$

on each boundary. Such a chiral boson is familiar from the discussion of the correspondence between pure $U(1)$ Chern-Simons theory on a three-dimensional manifold and the chiral boson RCFT on its boundary [21]. To better understand what happens, it is useful to expand $\widetilde{\varphi}(x^+)$ in Fourier modes:

$$\widetilde{\varphi}(x^+) = \widetilde{\varphi}_0 + \sum_{n \neq 0} \widetilde{\varphi}_n e^{inx^+} \, . \tag{25}$$

All modes of $\widetilde{\varphi}$ except for the zero mode[8] can be reconstructed from the modes of the current $j(x^+)$. However, it is only the zero mode that can carry electrical charge; indeed, from the $jj$ OPE we formally deduce that

$$\widetilde{\varphi}(z)j(0) \sim \frac{1}{z} \quad \Rightarrow \quad [\widetilde{\varphi}_n, j_0] = \delta_{n,0} \, . \tag{26}$$

An important issue is whether the zero mode $\widetilde{\varphi}_0$ is physical, which will only be true if it corresponds to a gauge-invariant quantity in the bulk. In the case of a single-sided geometry, the expectation value of $\widetilde{\varphi}_0$ can be shifted by a constant gauge transformation in the bulk, which does not modify at all the physical data contained in $\langle j_+(x^+) \rangle$. Thus, in this case the zero mode is unphysical and all the data we need to reconstruct the bulk field is encoded in

---

[8]While the concept of "zero mode" of a chiral object is not quite well-defined, we only use this terminology as an intermediate step to understanding what happens in the case of two boundaries.

the boundary current; indeed, we can easily check that the expression for the operators (19) that make up $\hat{\phi}$ does not involve the zero mode

$$\widetilde{\varphi}(\widehat{x}) - \widetilde{\varphi}(x) = \frac{2}{k} \int_x^{\widehat{x}} j(x^+) \,. \tag{27}$$

In the case of two boundaries, the expression for $\hat{\phi}$ contains a contribution from the Wilson line

$$W_{LR}(\widehat{x}_L, \widehat{x}_R) = e^{iq[\widetilde{\varphi}(\widehat{x}_R) - \widetilde{\varphi}(\widehat{x}_L)]} \,. \tag{28}$$

The zero mode $\widetilde{\varphi}_0^L - \widetilde{\varphi}_0^R$ of the (unexponentiated) Wilson line cannot be rewritten in terms of the boundary currents. However, while the zero modes $\widetilde{\varphi}_{L/R}^0$ are not separately gauge invariant, their difference cannot be changed by a gauge transformation, and thus is physical. Thus, the Wilson line (which did not exist in the single-boundary case) is now a physical operator acting on the Hilbert space of the two CFTs, and its charge is carried by the zero mode.

The expression (28) indicates that the Wilson line behaves as a vertex operator associated to a *non-chiral* free boson

$$\varphi(x_L^+, x_R^+) = \widetilde{\varphi}(x_R^+) - \widetilde{\varphi}(x_L^+) \,, \tag{29}$$

whose left-moving part originates from the CFT on the left boundary and right-moving part - from the CFT on the right boundary[9], with a shared zero mode. This is precisely what happens in the case of pure Chern-Simons theory on a manifold with two boundaries (the annulus), where the chiral bosons from the two boundaries combine into a single non-chiral boson [22]. Note however that at the microscopic level, the situation we have at hand is quite different from that of pure Chern-Simons theory: for us, Chern-Simons is just the low-energy limit of a consistent theory of quantum gravity in AdS$_3$ dual to some large $N$ CFT$_2$, which contains many additional degrees of freedom. This leads to differences in both the single-sided and the two-sided case.

In the duality of pure $U(1)$ Chern-Simons theory on a disk (i.e. global AdS$_3$) with the chiral boson, magnetic vortices in the Chern-Simons theory correspond to winding states of the chiral boson, with energy of order the Chern-Simons level, $k$. Such high energy states (recall that $k \sim N$ for weakly coupled Chern-Simons in the bulk) in the AdS bulk theory will no longer be well approximated by decoupled Chern-Simons, and the spectrum of winding states in our situation will be determined by details of the bulk physics.

In the two-sided case, the full microscopic Hilbert space is the tensor product of the CFT Hilbert spaces on the left and the right boundary, and it has a very different structure from that of the non-chiral compact boson CFT dual to pure Chern-Simons on a spacetime with two boundaries. In the latter case, due to the zero mode, there is no natural way to split the Wilson line in pure Chern-Simons theory into a left- and a right-boundary contribution, and thus the Hilbert space does not factorize. The same conclusion applies to the Wilson line we found perturbatively around the eternal black hole background.

The fact that the zero mode of $\varphi$ can only be defined in two-sided geometries, in addition to the non-existence of fixed CFT operators whose product gives the Wilson line, suggests that the latter is a state-dependent operator. Note that at low energies, the Wilson line will behave as the exponential of the non-chiral boson (29) around *any* state dual to a two-sided geometry, including states dual to spacetimes with long wormholes [23]. In particular, it behaves as if it were a primary chiral vertex operator $e^{iq\widetilde{\varphi}_{L/R}(x_{L/R}^+)}$ from the point of view of the CFT on the left/right boundary, i.e. it behaves as a *local* operator from the point of view of either CFT. This follows simply from the bulk operator algebra.

---

[9]The coordinate $x_R^+$ is a *right-moving* coordinate in the right CFT, due to the opposite orientation of the right boundary with respect to the radial direction in the bulk.

The commutation relations of the Wilson line with the low-lying CFT operators can be deduced from the relevant Dirac brackets in bulk perturbation theory. We perform this analysis in the next section.

## 2.3 Choice of gauge and quantization

In the previous section, we showed that an essential ingredient of the bulk field $\hat{\phi}$ is the boundary-to-boundary Wilson line, a pure-gauge configuration that only exists on manifolds with two boundaries and is charged under $Q = \frac{1}{2}(Q_R - Q_L)$. The purpose of this section is to work out the Dirac brackets of the Wilson line with the gauge-invariant bulk fields, from which the commutation relations of the Wilson line operator with the low-lying CFT operators follow. While the end result could have simply been inferred from the commutators of the currents and the definition (24), we use this technically simple example to illustrate how the computation would proceed in general and to outline the main physical issues that arise.

The computation of the commutators proceeds in three steps:

1. **Fix a gauge**. In order to obtain the correct commutators, in particular that of the Wilson line with $Q_{L/R}$, it is essential to perform a careful treatment of the choice of gauge on a manifold with two asymptotic boundaries. The choice of gauge condition should not restrict the boundary data, but at the same time it should completely determine the bulk gauge field in terms of it.

2. **Compute the Dirac brackets** of the gauge-fixed bulk fields.

3. **Express the bulk fields in terms of boundary operators** using the boundary-to-bulk dictionary, and deduce the corresponding boundary commutators.

Let us start by discussing the choice of gauge. The usual gauge used in holography is radial/holographic gauge, which has the advantage that the expression for the bulk gauge field is *local* in the boundary currents[10]. Working out the Dirac brackets in this gauge, all components of the gauge field turn out to be neutral under the boundary charge. This matches well with the fact that in e.g. global AdS, there cannot exist any charged pure gauge field configurations.

However, in the eternal black hole, global radial gauge is too restrictive: first, it forbids the Wilson line, including its zero mode that we in principle would like to take on arbitrary values; secondly, it disallows two sets of independent boundary currents. This is particularly easy to see in the case of three-dimensional Chern-Simons theory, where the analysis is highly simplified by the fact that the Chern-Simons action is topological. Thus, one can replace the eternal black hole background by just flat space

$$ds^2 = dz^2 + 2dx^+ dx^- \tag{30}$$

with two boundaries, which we take to be at radial positions $z = 0$ and $z = a$.

As discussed, on-shell we have $A = d\lambda$. We would like to impose $A_- = 0$ at both boundaries; this leaves $\lambda(x^+, z)$. Moreover, we would like to impose that

$$\partial_+ \lambda(x^+, 0) = \frac{2}{k} \langle j_+^L(x^+) \rangle \,, \qquad \partial_+ \lambda(x^+, a) = \frac{2}{k} \langle j_+^R(x^+) \rangle \,. \tag{31}$$

Since we want $j_{L,R}(x^+)$ to be completely independent, it is clear that radial gauge, $A_z = 0$, is not an option, since then $\lambda = \lambda(x^+)$ only, which implies that the variations of the two boundary

---

[10]In non-radial gauges, e.g. the AdS analogue of Coulomb gauge [11] or the gauge (32) we use below, one finds expressions for the bulk gauge field that are explicitly non-local in the boundary currents.

currents are correlated. Let us try instead the gauge

$$\partial_z A_z = 0 \quad \Rightarrow \quad \lambda(x^+, z) = \lambda_L(x^+) + \frac{z}{a}(\lambda_R(x^+) - \lambda_L(x^+)). \tag{32}$$

As we see, this gauge condition allows us to have the boundary conditions we want, while completely fixing the gauge field everywhere in terms of the boundary data $j_{L,R}(x^+)$ and the zero mode of $\varphi \equiv \widetilde{\varphi}_R - \widetilde{\varphi}_L$ which, as we argued in the introduction, needs to be independently specified:

$$\frac{k}{2} A_+(x^+, z) = j_L(x^+) + \frac{z}{a}(j_R(x^+) - j_L(x^+)), \tag{33}$$

$$A_z(x^+, z) = \frac{1}{a}\varphi(x^+). \tag{34}$$

The non-chiral boson $\varphi(x_L^+, x_R^+)$, which *a priory* depends on two sets of lightlike boundary coordinates $x_{L,R}^+$, satisfies

$$\frac{k}{2} \partial_{x_L^+} \varphi(x_L^+, x_R^+) = -j_L(x_L^+), \quad \frac{k}{2} \partial_{x_R^+} \varphi(x_L^+, x_R^+) = j_R(x_L^+), \tag{35}$$

where it is self-understood that $j_{L,R}$ only have a + component. As already explained, $x_L^+$ is a left-moving coordinate on the left boundary, but $x_R^+$ is a right-moving coordinate on the right boundary. In (34) we have taken $x_L^+ = x_R^+ = x^+$, which is why only one argument appears. Note also that $A_z$ is non-locally determined in terms of the boundary currents. This seems to be a generic feature of non-radial gauges.

Once we have fixed the gauge, we can now work out the Dirac brackets of the remaining degrees of freedom in this gauge. This is done in appendix A, and we find

$$\{A_+(x^+, z), A_+(x'^+, z')\}_{D.B.} = -\frac{4\pi}{k} \partial_+ \delta(x^+ - x'^+)\left(1 - \frac{z + z'}{a}\right), \tag{36}$$

$$\{A_+(x^+, z), A_z(x'^+, z')\}_{D.B.} = -\frac{4\pi}{ka} \delta(x^+ - x'^+). \tag{37}$$

The first Dirac bracket is perfectly consistent with the expression (33) for $A_+$ in terms of the boundary currents and the current commutator. The second Dirac bracket tells us the commutator of the field $\varphi$ with the boundary currents:

$$\{j_L(x^+), \varphi(x'^+)\}_{D.B.} = \{j_R(x^+), \varphi(x'^+)\}_{D.B.} = -2\pi\delta(x^+ - x'^+). \tag{38}$$

It is easy to check, using these expressions, that the Wilson line has the correct commutators with the boundary charges.

One can also work out the Dirac brackets of the charged scalar $\hat{\phi}$ with $\varphi$ and check that the charge of a bulk scalar framed to one of the boundaries is correctly rendered. See appendix A for details. Since the Chern-Simons action is topological, the commutation relations that we derived are valid not only in the eternal black hole background, but also in any three-dimensional space-time with two boundaries.

The same computation can in principle be performed in higher dimensions. On the one hand, the analysis is complicated by the fact that we now need to work on the actual black hole background, since the action is no longer topological. On the other hand, for Maxwell theory the CFT operators are given by the boundary limit of only gauge-invariant bulk quantities, whose Dirac brackets can be computed without explicitly solving the gauge condition, as we show in section 3.3.

## 3 Charged scalar coupled to Maxwell theory in $D > 3$

In this section, we would like to show that the same analysis can be performed for Maxwell theory in $D > 3$. The results are qualitatively the same, even though the details change and, unfortunately, in this case we will not have nice, explicit expressions as in $D = 3$.

As in the previous section, we start with an analysis of the bulk equations of motion, and show they require the inclusion of the Wilson line in the expression for $\hat{\phi}$. Unlike in three dimensions, the shape of the Wilson line now does matter, even in the small coupling limit. In section 3.2, we sketch the computation of the value of a nicely-shaped (unexponentiated) Wilson line in terms of the boundary currents and the relative zero mode of the boundary gauge parameters. In 3.3, we show that even without knowing the explicit expression for the Wilson line in terms of the boundary currents, its commutators with local operators on the two boundaries are local, in the sense that they vanish outside the boundary lightcone.

### 3.1 Equations of motion analysis

Consider now the action

$$S = \int d^D y \sqrt{g} \left( -\frac{1}{4e^2} F_{\mu\nu} F^{\mu\nu} - D_\mu \phi \, D^\mu \phi^\star - m^2 |\phi|^2 \right), \tag{39}$$

where $D_\mu = \partial_\mu - iqA_\mu$, with $q \in \mathbb{Z}$, and $D = d + 1$. The equations of motion read

$$\nabla_\mu F^{\mu\nu} = e^2 J^\nu, \qquad (\Box - m^2)\phi = iq(\phi \nabla_\mu A^\mu + 2A^\mu \partial_\mu \phi) + q^2 A^2 \phi. \tag{40}$$

In the limit $e \to 0$, we can neglect the backreaction of the scalar field on $F_{\mu\nu}$, and the equation for $\phi$ becomes linear (in $\phi$). This limit allows us to consistently include all contributions to the charge, while still having manageable equations.

In this approximation, the scalar equation can be written entirely in terms of the gauge-invariant quantities $\hat{\phi}(y)$, defined in (1), and the field strength

$$(\Box - m^2)\hat{\phi} = -iq \, \hat{\phi} \, \nabla^M \int_\Gamma F_{MP} dy^P - 2iq \, \nabla^M \hat{\phi} \int_\Gamma F_{MP} dy^P + q^2 \, \hat{\phi} \, g^{MN} \int_\Gamma F_{MP} dy^P \int_\Gamma F_{NQ} dy^Q, \tag{41}$$

where the integral is performed the path $\Gamma$ that appears in the definition of $\hat{\phi}$ and runs from the bulk point $y$ to the boundary point $\hat{x}_R$. In deriving this expression, we have used the identity[11]

$$\partial_M \int_\Gamma A = -A_M(y) - \int_\Gamma F_{MP} dx^P. \tag{42}$$

Note that, unlike in three dimensions where $F = 0$, in $D > 3$ the shape of the Wilson line does matter. The expression for $\hat{\phi}$ can be obtained by solving (41) perturbatively in the field strength $F$. At zeroth order $F = 0$, so $A = d\lambda$. Then, $\hat{\phi}$ satisfies the free wave equation and the solution is

$$\hat{\phi}^{(0)}(y) = \int d^d x_L K_L(y|x_L) \, \hat{\mathcal{O}}_L(x_L) + \int d^d x_R K_R(y|x_R) \, \hat{\mathcal{O}}_R(x_R), \tag{43}$$

where

$$\hat{\mathcal{O}}_L(x_L, \hat{x}_R) = \mathcal{O}_L(x_L) \, e^{iq(\lambda(\hat{x}_R) - \lambda(x_L))}, \qquad \hat{\mathcal{O}}_R(x_R, \hat{x}_R) = \mathcal{O}_R(x_R) \, e^{iq(\lambda(\hat{x}_R) - \lambda(x_R))}. \tag{44}$$

---

[11] To evaluate the derivative of the Wilson line, it is useful to work in coordinates in which the Wilson line stretches along $z$, at $x^\mu = const$, where $z$, $x^\mu$ become approximately Poincaré coordinates near the boundary. In $D > 3$, the normalizable boundary condition has $\lim_{z \to 0} A_\mu = 0$.

Near the boundary in Poincaré coordinates, the asymptotic behaviour of the gauge field is

$$A_\mu(x,z) \sim z^{d-2} a_\mu(x), \qquad A_z \sim z^{d-3} a_z(x), \tag{45}$$

assuming normalizable boundary conditions. This implies that near each boundary, the allowed gauge parameters take the form

$$\lambda = \lambda_0 + f(x) z^{d-2} + \dots, \tag{46}$$

where $\lambda_0$ is a number, $\partial_\mu \lambda_0 = 0$. This implies that $\hat{\mathscr{O}}_R = \mathscr{O}_R$, whereas $\hat{\mathscr{O}}_L = e^{iq(\lambda_0^R - \lambda_0^L)} \mathscr{O}_L$. As already discussed, the zero mode of the relative gauge parameter carries charge, and this is already the most non-trivial part of the Wilson line.

Next, we can pertubatively include the contribution of the integrated field strengths. These contributions will be entirely expressible in terms boundary currents, since $F_{MN}$ satisfies second order equations of motion and its boundary values are determined by the CFT currents via

$$\lim_{z \to 0} \sqrt{g} F^{z\mu} = j^\mu. \tag{47}$$

Thus, the integrated field strengths will just dress the operators and the Wilson line by additional powers of the CFT currents. The expression one obtains at the end is precisely of the form (3).

We can also derive (3) from the known fact [8] that in radial gauge, the expression for $\hat{\phi}$ is given only by smearing dressed operators $\mathscr{O}^{(j)}$, for some dressing by the current. Let us rename the path that unites the point $y$ - where the bulk field is inserted - to the boundary point $\hat{x}_R$ to be $\Gamma_R$, shown in figure 6(a). Since in the approximation in which we are working, the equation of motion (41) is linear in $\hat{\phi}$, the solution for the bulk field $\hat{\phi}$ in the interior of the black hole consists of two pieces

$$\hat{\phi}(\Gamma_R) = \hat{\phi}_L(\Gamma_R) + \hat{\phi}_R(\Gamma_R), \tag{48}$$

where $\hat{\phi}_L(\Gamma_R)$ only has support on the left boundary and $\hat{\phi}_R(\Gamma_R)$ only on the right one, but each of them has charges $Q_L = 0$, $Q_R = q$ and separately solves the wave equation with $\Gamma = \Gamma_R$, as we have explicitly indicated. The idea is now to evaluate $\hat{\phi}_{L/R}$ separately using radial gauge. However, as we already discussed, global radial gauge is not allowed in the eternal black hole background; instead, we will be imposing radial gauge patchwise in the left/right parts of the space-time (which include the interior bulk point all the way to the left/right boundary) and then put the results together. Our procedure is depicted in figure 4.

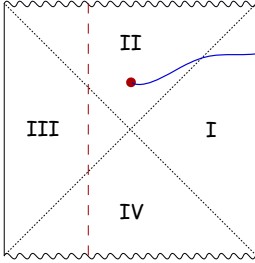

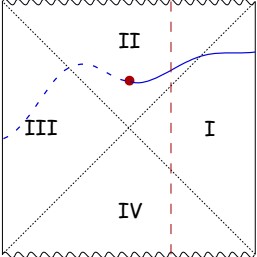

(a) We can find the boundary expression for $\hat{\phi}_R$ by imposing radial gauge to the right of the dotted line in the figure above.

(b) By imposing radial gauge to the left of the vertical dotted line, we find the boundary expression for $\hat{\phi}_L(\Gamma_L)$, which is framed to the left via the dashed Wilson line.

Figure 4: Argument to find the boundary representation of $\hat{\phi}$ using patchwise radial gauge.

We first impose radial gauge to the right of the dotted line in figure 6(a). Since the Wilson line attached to $\hat{\phi}$ ends on the right boundary, we know that $\hat{\phi}_R$ can be written as some specific smearing over dressed operators $\mathcal{O}_R^{(j)}$, whose precise dressing depends on $\Gamma_R$. This corresponds to the first term in (3).

As for $\hat{\phi}_L$, we now impose radial gauge in the left half of the eternal black hole (figure 6(b)). If $\hat{\phi}_L$ were framed to some point on the left boundary, say via a curve $\Gamma = -\Gamma_L$, then it would have some specific expression in terms of dressed operators on the left boundary involving $\mathcal{O}_L^{(j)}$ - where, again, the precise dressing depends on the shape of $\Gamma_L$ and on its boundary endpoint. We denote this left-framed operator by $\hat{\phi}_L(\Gamma_L)$, which satisfies (41) with $\Gamma = -\Gamma_L$. However, $\hat{\phi}_L(\Gamma_R)$ is framed to the right boundary, and not the left, so the expression we want differs from the expression for $\hat{\phi}_L(\Gamma_L)$ precisely by a boundary-to-boundary Wilson line stretching along $\Gamma_L + \Gamma_R$:

$$\hat{\phi}_L = \phi_L(\Gamma_L) \cdot W_{LR}(\Gamma) \,, \qquad W_{LR} = \exp\left( iq \int_{\Gamma_L} A + iq \int_{\Gamma_R} A \right) . \qquad (49)$$

This represents the second term in (3). Using the equation of motion for $\hat{\phi}_L(\Gamma_L)$, it is not hard to show that, irrespectively of how we choose $\Gamma_L$, $\hat{\phi}_L$ satisfies (41) with $\Gamma = \Gamma_R$. This shows how the shape of the Wilson line is constrained by the equations of motion. Of course, in general the Wilson line need not pass through the bulk point $y$; changing its shape will simply multiply the expression for the bulk field by $e^{iq \oint A}$, where the integral is performed along the closed contour corresponding to the difference of the two Wilson lines. Converting the contour integral to a surface integral over the field strength, the difference in Wilson lines is a functional of the boundary currents only.

## 3.2 Evaluating the Wilson line

In the above section, we have established the necessity of the Wilson line also in higher dimensions. Its most non-trivial part - which is not encoded in the CFT currents - is the zero mode, which we have already discussed; however, as we are mostly interested in the *localized* Wilson line, it would be very interesting to also have an expression for its non-zero modes in terms of the CFT currents.

Unlike in three dimensions, where the relation between the CFT currents and the Wilson line is very simple (35), here we will unfortunately be unable to provide completely explicit expressions for the Wilson line in terms of the currents. We will, however, describe in detail the procedure through which such an expression may be obtained. We write the final result in terms of integrals over the bulk-to-boundary propagator in AdS-Schwarzschild, which is known numerically (see e.g. [24]) and can be used in principle to compute the Wilson line.

For simplicity, we work with the unexponentiated Wilson line, $\varphi = \int_\Gamma A$. To determine the value of $\varphi$, we must first pick a shape. We concentrate on the planar AdS-Schwarzschild black brane, though very similar statements hold for the spherically symmetric black hole. The metric of the $\text{AdS}_{d+1}$-Schwarzschild black brane is

$$ds^2 = -f(r)dt^2 + \frac{dr^2}{f(r)} + r^2 d\vec{x}^2 \,, \qquad f(r) = \frac{r^2}{\ell^2} - \frac{\mu}{r^{d-2}} \,, \qquad (50)$$

where $\ell$ is the AdS length and $\mu$ parametrizes the mass. This set of coordinates is only valid in region I of the eternal black hole, but we can use similar coordinates in each of the four regions. In region III, the coordinate $t$ runs in the opposite direction from region I.

We would like to choose a nice family of Wilson lines in this geometry. A natural and simple choice are Wilson lines that stretch along bulk geodesics that unite points of $t_L = -t_0, t_R = t_0$ on the two boundaries and stay at $\vec{x} = const$.

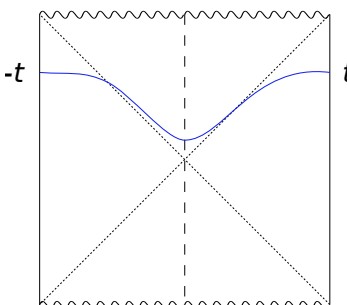

Figure 5: Wilson line stretching along a boundary-to-boundary geodesic.

These geodesics are labeled by the conserved "energy" $E$. They are only non-trivial on the $(t, r)$ plane, where they satisfy

$$f(r)\dot{t} = E, \qquad \dot{r}^2 = f(r) + E^2. \tag{51}$$

Here $\dot{} = d/d\sigma$, where $\sigma \in (-\infty, \infty)$ is the affine parameter along the geodesics. We choose the origin for $\sigma$ such that $\sigma = 0$ at $t = 0$ in region II. We obtain the full geodesic by gluing the solution across the three regions. Instead of $E$, we can alternatively parametrize the geodesics by the time $t_0$ they reach on the right boundary; this is shown explicitly in appendix B for the case of three bulk dimensions. The geodesic with $t_0 = 0$ is the one that goes straight through the bifurcation surface.

Note that these geodesics also provide a global time foliation of the eternal black hole. Indeed, introducing a timelike coordinate $\tau$ such that the geodesics are lines of constant $\tau$, the metric can be written as

$$ds^2 = d\sigma^2 - a^2(\sigma, \tau)d\tau^2 + b^2(\sigma, \tau)dx_i^2, \quad i = 1, \ldots, d-1. \tag{52}$$

Note that $\tau$ must equal $\pm t$ as we approach the boundaries at $\sigma \to \pm\infty$. We give explicit change of coordinates from $(r, t)$ to $(\sigma, \tau)$ for the special case of three dimensions in appendix B.

We would now like to compute the value of the Wilson line stretching along these geodesics. Unlike the general Wilson line - which is labeled by two different boundary positions - these symmetric Wilson line can be labeled just by their endpoint $(t, \vec{x})$ on the right boundary

$$\varphi(t, \vec{x}) = \int_{-\infty}^{\infty} A_\sigma(\tau, \sigma, \vec{x})\, d\sigma, \tag{53}$$

where the integral is performed along a line of constant $\tau, \vec{x}$, with $\lim_{\sigma \to \infty} \tau = t$.

Next, we need the expression for $A_\sigma$ all along the geodesic. For this, we first need to fix an allowed gauge in the bulk, e.g. $\partial_\sigma A_\sigma = 0$, that completely determines $A_\sigma$ in terms of the boundary currents and the zero mode, just like in the three-dimensional analysis of section 2.3. However, solving the gauge condition is extremely tedious on the black hole background.

It turns out to be much simpler to work out the derivatives of the Wilson line with respect to the boundary coordinates $t, \vec{x}$, as these only involve integrals of the gauge-invariant field strength. Consider first the derivative of $\varphi(t, \vec{x})$ with respect to the boundary time $t$, which is given by the difference as $\Delta t \to 0$ between two bulk geodesics with endpoints at $t$ and $t + \Delta t$. In the coordinates (52), we have

$$\Delta\varphi(t, \vec{x}) = \int_{\tau=const.} d\sigma\, F_{\tau\sigma}(\tau, \sigma, \vec{x})\,\Delta t. \tag{54}$$

Written covariantly, we have

$$\partial_t \varphi(t, \vec{x}) = \int n^M F_{MN} t^N d\sigma \,, \tag{55}$$

where $t^M = \frac{\partial y^M}{\partial \sigma}$ is the tangent vector to the geodesic, whereas $n^M = \frac{\partial y^M}{\partial \tau}$ is the deviation vector between the two neighbouring geodesics.[12] It is useful to work in terms of the Schwarzschild coordinates $(t, r)$, patching them together as needed. Then,

$$\partial_t \varphi(t, \vec{x}) = \int F_{tr} \left( \frac{\partial t}{\partial \tau} \frac{\partial r}{\partial \sigma} - \frac{\partial r}{\partial \tau} \frac{\partial t}{\partial \sigma} \right) d\sigma = \int F_{tr} \, a(\sigma, \tau) d\sigma \,. \tag{56}$$

In turn, $F_{rt}$ is entirely determined by the CFT currents via the following second-order equation[13]

$$\left( -\frac{r^2}{f(r)} \partial_t^2 + r^2 f(r) \partial_r^2 + \partial_i^2 \right) F_{rt} + (r^2 f' + (d+1)rf) \partial_r F_{rt} + (d-1)(f + rf') F_{rt} = 0 \tag{58}$$

and the boundary conditions

$$\lim_{r \to \infty} r^{d-1} F_{rt} = j^0_{L/R} \tag{59}$$

near the left/right boundary. This implies that all along the geodesic, the solution for $F_{rt}$ can be written as

$$F_{rt}(y) = \int d^d x_L \, K_L^{(F)}(y|x_L) \, j^0_L(x_L) + \int d^d x_R \, K_R^{(F)}(y|x_R) \, j^0_R(x_R) \,, \tag{60}$$

where $K_{L,R}^{(F)}$ satisfy the equation of motion (58) and $y = (t, r, \vec{x})$. In pure AdS, these propagators are simple derivatives of delta functions; unfortunately, on a general eternal black hole background, the expressions for them are not known. Using these ingredients, the final expression for the Wilson line will take the form

$$\partial_t \varphi(x) = \int d^d x' \, \mathcal{K}_L(x|x') \, j^0_L(x') + \mathcal{K}_R(x|x') \, j^0_R(x') \,, \tag{61}$$

where $x$ denotes all the boundary coordinates. It would be extremely interesting to compute the smearing functions $\mathcal{K}_{L,R}$ and see whether, as in three dimensions, the derivative of the unexponentiated Wilson line is given by a simple expression in terms of the boundary currents.

The above gives the derivative of the unexponentiated Wilson line with respect to $t$. We can similarly compute its derivative with respect to $x^i$ by integrating the corresponding field strength

$$\partial_i \varphi(t, \vec{x}) = \int r F_{i\sigma} d\sigma = \int \left( r F_{ir} \frac{\partial r}{\partial \sigma} + r F_{it} \frac{\partial t}{\partial \sigma} \right) d\sigma \,. \tag{62}$$

In empty AdS, $F_{ir}$ satisfies a decoupled second order differential equation, which can be used to determine it everywhere in terms of $j^i_{L,R}$. However, in a black hole background, there is

---

[12]This vector field can be determined from the geodesic deviation equation and the boundary conditions $n^M = \pm \delta_0^M$ as $\sigma \to \pm \infty$.

[13]This is quite similar (but not exactly the same when the black hole is present) to the wave equation for a scalar field

$$r^2 (\Box - m^2) \Phi = \left( -\frac{r^2}{f(r)} \partial_t^2 + r^2 f(r) \partial_r^2 + \partial_i^2 \right) \Phi + (2rf + r^2 f') \partial_r \Phi - m^2 r^2 \Phi \,. \tag{57}$$

In fact, $F_{rt}$ behaves near infinity just as $r\Phi$ with $m^2 = -2(d-2)/\ell^2$, which is inside the BF bound in $AdS_{d+1}$.

a mixing with $F_{it}$, which is determined by both $j^0$ and $j^i$. Again, it would be extremely interesting if this expression could be evaluated exactly, to find out whether $\partial_i \varphi$ bears a simple relation to the CFT currents.

Thus, we can in principle find all derivatives of the (unexponentiated) Wilson line as a linear functions of the currents. The only missing piece from the full Wilson line is the zero mode, which can be added in by hand.

## 3.3 Locality of the Wilson line

While the expressions derived in the previous section show how to obtain, in principle, an expression for the unexponentiated Wilson line in terms of the boundary currents and the additional zero mode, they are not very useful for understanding the behaviour of the Wilson line within correlators. In this section we show that, with a particular choice of the path, the Wilson line behaves as a local operator from the point of view of either CFT, in the sense that it commutes with all local CFT operators at spacelike separation.

We would thus like to compute the commutator of a low-lying local CFT operator $\mathscr{A}(t', \vec{x}')$ with Wilson line at some (earlier) global time $\tau$, where $\mathscr{A}$ is either a charged operator or a current. For this, it is sufficient to know the commutator of $\mathscr{A}(\tau', \vec{x}')$ with the unexponentiated Wilson line $\varphi(\tau, \vec{x})$, which can be obtained by evaluating the corresponding bulk Dirac bracket. While it is easy to compute equal-time commutators in the bulk, non-equal time ones are much harder. Our strategy will be to first use backward time evolution in the bulk to write $\mathscr{A}(t', \vec{x}')$ in terms of its value and first derivative on a $\tau = const.$ surface, and then use the equal-time Dirac brackets to compute the bulk commutators.

Remember the Dirac brackets are defined in terms of the Poisson brackets via

$$\{\mathscr{O}_1, \mathscr{O}_2\}_{D.B.} = \{\mathscr{O}_1, \mathscr{O}_2\}_{P.B.} - \{\mathscr{O}_1, \chi_i\}_{P.B.}(C^{-1})^{ij}\{\chi_j, \mathscr{O}_2\}_{P.B.}, \tag{63}$$

where $\chi_i$ represent the second class constraints and $C_{ij} = \{\chi_i, \chi_j\}_{P.B.}$. The main simplification we will use in this section is that Dirac brackets of gauge-invariant quantities (as opposed to the Dirac brackets of non-gauge-invariant fields that have been gauge fixed) can be computed without explicitly solving for $C^{-1}$, and in fact they just equal the Poisson brackets.[14]

We consider quantising Maxwell theory on a manifold with metric (52), in the gauge $\partial_\sigma A_\sigma = 0$. The momenta conjugate to the gauge field are

$$\pi^a = F^{a\tau}\sqrt{g}, \tag{64}$$

where the index $a$ runs over all the spatial directions in the bulk. The constraints read

$$\chi_1 = \pi^0, \quad \chi_2 = \partial_a \pi^a - J^0\sqrt{g}, \quad \chi_3 = \partial_\sigma A_\sigma, \quad \chi_4 = \partial_\sigma \pi^\sigma \frac{a^2(\sigma, \tau)}{\sqrt{g}} + \partial_\sigma^2 A_0, \tag{65}$$

where $\chi_4$ is supposed to implement $\partial_\tau \partial_\sigma A_\sigma = 0$. Using the above, one can easily show that the equal time Dirac bracket of the gauge field strength with the Wilson line is simply equal to their Poisson bracket. The equal-time Dirac bracket of the scalar with the Wilson line is zero.

Let us start by computing the commutator of $F_{\sigma\tau}(\tau', \sigma', \vec{x}')$ with a (geodesic) Wilson line $\varphi(\tau, \vec{x}) = \int A_\sigma(\tau, \sigma, \vec{x})d\sigma$, for $\tau' > \tau$. This component of the gauge field strength satisfies a second order differential equation by itself. If the bulk point $y'$ is deep in the interior, then

---

[14]For Maxwell theory on a manifold without boundary, this can be simply be understood from the fact that before imposing the gauge-fixing conditions, the constraints $\chi_{1,2}$ below used to be first class constraints that generate gauge transformations, and thus have zero Poisson bracket with the gauge-invariant quantities of interest. Thus, the only non-zero correction to the Poisson bracket can come if $(C^{-1})^{34}$ is non-zero; however, one can explicitly check that it vanishes, using the fact that $\{\chi_1, \chi_2\}_{P.B.} = 0$.

we can use the initial conditions for $F_{\sigma\tau}$ and its first derivative on the $\tau = const.$ surface to construct $F_{\sigma\tau}(\tau', \sigma', \vec{x}')$

$$F_{\sigma\tau}(\tau', \sigma', \vec{x}') = \int d^{d-1}x'' d\sigma'' \Big[ G_F(\tau', \sigma', \vec{x}' | \tau, \sigma'', \vec{x}'') F_{\sigma\tau}(\tau, \sigma'', \vec{x}'') +$$
$$+ G_{F'}(\tau', \sigma', \vec{x}' | \tau, \sigma'', \vec{x}'') \partial_\tau F_{\sigma\tau}(\tau, \sigma'', \vec{x}'') \Big] , \quad (66)$$

where $G_F, G_{F'}$ are determined from the equations of motion and only have support inside the intersection of the lightcone emanating from $y'$ with the $\tau = const.$ surface. It is not hard to show that the second term has vanishing Poisson bracket with $A_\sigma$, and thus can be dropped in computing the commutator with the Wilson line. Finally, we obtain

$$\{F_{\sigma\tau}(\tau', \sigma', \vec{x}'), \varphi(\tau, \vec{x})\} = \int d\sigma\, G_F(\tau', \sigma', \vec{x}' | \tau, \sigma, \vec{x}) . \quad (67)$$

We are ultimately interested in the case in which the insertion of the field strength is on the boundary, since $\lim_{\sigma' \to \pm\infty} \pi^\sigma = j^0_{L/R}$. Then, $G_F$ is not only fixed by the data on the $\tau = const.$ surface, but also by requiring normalizable boundary conditions as $|\sigma| \to \infty$. Thus, we have reduced computing the Dirac bracket of the boundary current with the Wilson line to evaluating the simple expression (67). When $\tau' = \tau$, it is easy to see this expression yields the expected commutator

$$\{j^\mu_{L/R}(\tau, \vec{x}'), \varphi(\tau, \vec{x})\} = \mp \delta^\mu_0 \delta^{(d-1)}(\vec{x} - \vec{x}') \quad (68)$$

with the CFT currents, and thus the charges. When $\tau' \neq \tau$, there are also nontrivial commutators of $j^i$ and the Wilson line.

Since in (67) all contributions come from inside the bulk lightcone associated to $(\tau', \sigma', \vec{x}')$, this implies (at least for the nicely-shaped Wilson lines we are considering) that the commutator of the Wilson line with the boundary operator is local, in the sense that it vansihes outside the boundary lightcone. It also implies that the commutator of the Wilson line with local operators on a single boundary only depends on the geometry outside the horizon, and thus will be the same in the eternal black hole, or in a long wormhole spacetime.

Finally, we can also compute the commutator of the Wilson line with the charged boundary operators. The equal-time Dirac bracket of the scalar field or its conjugate momentum with the Wilson line can be shown to be zero. In order to get a non-zero answer at unequal times, we need to take into account the non-linear evolution of the scalar in terms of both $\phi$ and $A$; terms proportional to the field strength on the initial surface will contribute to the commutator with the Wilson line operator. Given that all the propagators involved are causal, it is clear that the resulting answer will vanish outside the boundary lightcone; in this sense, the Wilson line behaves as a local operator from the CFT point of view. This conclusion holds not only in the eternal black hole, but also in any two-sided black hole spacetime.

## 4 CFT representation of the boundary-to-boundary Wilson line

In this section, we discuss the CFT representation of the boundary-to-boundary Wilson line. In the introduction, we have argued that the Wilson line should correspond to a state-dependent operator; in any case, it action on the small Hilbert space around a given state should be well-defined. The subject of the present section is to determine this action on the thermofield double state.

We will consider both the operator

$$\varphi = \int_\Gamma A_\mu dx^\mu \tag{69}$$

and the Wilson line

$$W = P \exp\left( iq \int_\Gamma A_\mu dx^\mu \right), \tag{70}$$

regulated by appropriate counterterms. Remember that the operator $\varphi$ is - strictly speaking - not a well-defined operator, as it is only defined mod $2\pi$; however, its action is much easier to evaluate than that of $W$. In the special case that the effective bulk theory is weakly coupled Chern-Simons in three dimensions, these operators will be independent of the path.

These operators cannot be expressed purely in terms of the boundary currents, however they do exist in CFT at least in $1/N$ perturbation theory, when acting on states that are small excitations of the thermofield double state. We will exhibit this by first determining their matrix elements between such states (which abstractly defines the operators), and then by constructing them as the leading divergence in the bulk operator product of left and right charged operators that approach the bifurcation surface. The latter construction also requires a particular state, since in general states, there will be no divergence in that operator product. Finally, we will discuss the relation of the Wilson line with the Papadodimas-Raju construction of the mirror operators in the eternal black hole.

## 4.1 Action on the thermofield double state

To specify the operators abstractly on the small Hilbert space, it is sufficient to determine their action on the black hole state and their commutation relations with left and right single trace operators. In this section, we concentrate on the action of $\varphi$ on the small Hilbert space associated to the thermofield double state, which can be entirely reconstructed from

$$\varphi |\Psi\rangle_{\text{tfd}}, \qquad [\varphi, \mathcal{O}] \dots |\Psi\rangle_{\text{tfd}}. \tag{71}$$

The latter can be determined order by order in bulk perturbation theory, as we have done in sections 2.3 and 3.3. The former can be found using the Euclidean path integral description of the thermofield state, as we show below.

Consider the path integral description of the thermofield double state. From the CFT perspective, it is given by the Euclidean path integral on an cylinder of length $\beta/2$, interpreted as a wavefunctional of boundary conditions on the two boundaries of the cylinder [25]. This gives an element of the tensor product Hilbert space.

We consider a bulk Wilson line stretching along the surface $t = 0$ at $\vec{x} = const$, i.e. the straight geodesic Wilson line $\varphi(0, \vec{x})$ that passes through the bifurcation point. Any other Wilson line can be obtained from this one by multiplication by a functional of the currents. The bulk dual is dominated by the saddle shown in figure 6. The Wilson line operator acting on this state is given by the associated insertion of $\int_{t=0} d\sigma A_\sigma$ in this bulk Euclidean path integral.

Consider deforming this Wilson line to run along the boundary. One has

$$\varphi(0, \vec{x}) = \int_0^{\beta/2} dt_E A_{t_E} + \int_B F, \tag{72}$$

where $B$ is the bulk euclidean slice at fixed $\vec{x}$. This operation only applies to the insertion of the operator $\varphi$ in this particular path integral, with no other insertions.

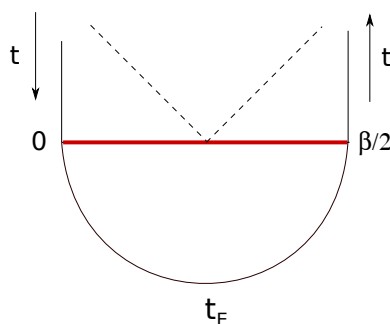

(a) Insertion in the path integral that corresponds to the Wilson line $\int_{t=0} d\sigma A_\sigma$.

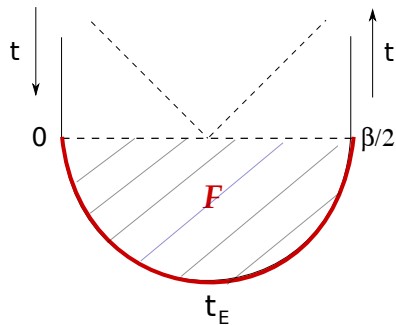

(b) This can be deformed to a Wilson line stretching along the Euclidean time direction at $r \to \infty$, by picking up a surface integral of the field strength.

Figure 6: Path integral evaluation of the Wilson line action on the thermofield double state.

In the three-dimensional weakly coupled Chern-Simons case, the field strength $F$ vanishes on the saddle point, and the above expression simplifies to

$$\varphi(0, x) = \frac{2}{k} \int_0^{\beta/2} dt_E \, j_{t_E}(it_E, x) \,, \tag{73}$$

where $j_{t_E}$ has been defined such that $j_{t_E}(i\beta/2, x) = i j_R^0(0, x)$ and $j_{t_E}(0, x) = -i j_L^0(0, x)$. The factors of $i$ come from the analytic continuation of the gauge field.

It is also interesting to consider the action of the Wilson line $\bar{\varphi}$ that still stretches along the $t = 0$ surface, but is smeared in the $x$ direction

$$\bar{\varphi} = \int dx \, \varphi(0, x) = \frac{2}{k} \int_0^{\beta/2} dt_E \int dx \, j_{t_E}(it_E, x) = \frac{2\pi i \beta}{k} Q \,, \tag{74}$$

where $Q = \frac{1}{2\pi i} \int dt_E \, j_{t_E}$ and we have used the fact that an insertion of $Q$ in the path integral that produces the thermofield double state is equivalent to an insertion of $Q_R$, or of $-Q_L$, since $(Q_L + Q_R)|\Psi\rangle_{\text{tfd}} = 0$. Thus, we find that the action of the spatially averaged Wilson line on the thermofield double state is given by

$$\bar{\varphi} |\Psi\rangle_{\text{tfd}} = \frac{2\pi i \beta}{k} Q |\Psi\rangle_{\text{tfd}} \,. \tag{75}$$

Note that the thermofield double state is *not* an eigenstate of $\bar{\varphi}$, as one may have naively expected. Thus, while the operator algebra of the unexponentiated Wilson line is that same as that of a free boson, its action on the thermofield double state is non-trivial. If we computed the action of $\varphi$ on other states dual to two-sided geometries, we expect that the answer would again be different, while the operator algebra stays the same.

We can perform a similar analysis in higher dimensions. There, the boundary contribution to the path integral vanishes due to the boundary conditions (45), but the bulk integral over $F$, which is gauge invariant, can be written in terms of the boundary currents operators ($j^0$) using the standard bulk to boundary kernel. This computation is a bit too complicated to perform here. However, it is easy to compute the action of the spatially averaged Wilson line $\bar{\varphi}$. This time we work with the spherically symmetric black hole geometry, whose metric is given by (50) with $d\vec{x}^2 \to d\Omega_{d-1}^2$, the metric on the unit $(d-1)$ sphere. We have

$$\bar{\varphi} = \int d^{d-1}\Omega \int_{r_+}^{\infty} dr \int_0^{\beta/2} dt_E F_{t_E r} . \tag{76}$$

In free Maxwell theory, the flux through a surface of radius $r$ at time $\tau$

$$\Phi(r, t_E) = \frac{1}{e^2} \int_{r, t_E = const.} d^{d-1}\Omega \, F^{t_E r} \sqrt{g} \tag{77}$$

is constant with respect to both $r$ and $t_E$ and it equals $iQ$. Using the metric (50), we find

$$\bar{\varphi} = e^2 \int_{r_+}^{\infty} \frac{dr}{r^2} \int_0^{\beta/2} dt_E \, \Phi = \frac{i\beta e^2}{2r_+} Q . \tag{78}$$

Again, we find a non-trivial action of the Wilson line on the thermofield double state

$$\bar{\varphi} |\Psi\rangle_{\text{tfd}} = \frac{i\beta e^2}{2r_+} Q |\Psi\rangle_{\text{tfd}} \tag{79}$$

proportional to the relative charge.

It is interesting to ask whether the expressions we found are consistent with the commutation relations of $\bar{\varphi}$ we found from the bulk analysis. In three dimensions, we have

$$[Q, \bar{\varphi}] = 2\pi i R , \tag{80}$$

where $2\pi R$ is the length of the spatial circle. The expectation value of this commutator is

$$2\pi i R = \langle \Psi_{\text{tfd}} | Q \bar{\varphi} | \Psi \rangle_{\text{tfd}} - \langle \Psi_{tfd} | \bar{\varphi} Q | \Psi \rangle_{\text{tfd}} = \frac{4\pi i \beta}{k} \langle \Psi_{\text{tfd}} | Q^2 | \Psi \rangle_{\text{tfd}} , \tag{81}$$

where we used the fact that $\bar{\varphi}$ is hermitean. Thus, for our calculation to be consistent, we should have

$$\langle \Psi_{\text{tfd}} | Q^2 | \Psi \rangle_{\text{tfd}} = \frac{kR}{2\beta} . \tag{82}$$

The expectation value of $Q^2$ can be computed, as in [16], by turning on an infinitesimal electric potential $\mu$ and computing the resulting expectation value of the relative charge $Q$

$$\langle Q \rangle_\mu = \frac{1}{Z} \sum_E q \, e^{-\beta(E-\mu q)} = \mu\beta \langle \Psi_{\text{tfd}} | Q^2 | \Psi \rangle_{\text{tfd}} + \mathcal{O}(\mu^2) . \tag{83}$$

In three-dimensional Chern-Simons theory, the electric potential is proportional to the $A_-$ component of the gauge field on the boundary, more precisely $A_- = -\mu$, as can be seen from (101). This in turn leads to a non-zero charge

$$Q_\mu = -\frac{k}{4\pi}\int dx\, A_- = \frac{kR\mu}{2}\,. \tag{84}$$

Comparing (84) and (83), we find perfect agreement with (82).

A similar comparison can be performed in higher dimensions. We have

$$[Q,\bar\varphi] = i\,\text{vol}(\Omega^{d-1})\,, \tag{85}$$

where $\text{vol}(\Omega^{d-1})$ is the volume of the $(d-1)s$ sphere. For consistency, we need that

$$\langle\Psi_{tfd}|Q^2|\Psi\rangle_{\text{tfd}} = \frac{\text{vol}(\Omega^{d-1})r_+}{\beta e^2}\,. \tag{86}$$

This agrees perfectly with the formulae in [16], who found $4\pi r_+/\beta e^2$ in four dimensions.

## 4.2  Construction via the bulk OPE

In this section we will describe a more physical way to construct the Wilson line operator, from the product of a pair of bulk operators with standard HKLL descriptions. This construction will still be state-dependent, in that it involves finding the operator coefficient of a certain divergence in the operator product which only exists around certain states (and in $1/N$ perturbation theory). Such a construction of the Wilson line operator as a limit of simple (i.e., products of a small number of single-traces) operators only applies to entangled states of the tensor product theory which are dual to black holes with a single bifurcation surface, in other words without a long throat.

Consider a gauge invariant charged field operator in the right wedge, dressed by a Wilson line that connects it to some point $\hat{x}_R$ on the right boundary. We denote it schematically by

$$\hat\phi_R(y) = e^{iq\int_y^{\hat{x}_R}}\phi(y)\,. \tag{87}$$

This has a standard HKLL description in terms of a perturbative expansion in integrals of products of single trace right CFT operators. Similarly, the corresponding anti-particle on the left, framed by a Wilson line to the left boundary, $\hat\phi_L^\dagger(y') = \phi^\dagger(y')e^{-iq\int_{y'}^{\hat{x}_L}}$, can be expressed entirely in terms of left operators.

Bringing these two operators together near the bifurcation surface results in a singular OPE, at least in $1/N$ perturbation theory, for states close to the thermofield state. This is just the bulk OPE. In particular, if one ignored issues of gauge invariance,

$$\phi(y)\phi^\dagger(y') \sim \frac{1}{|y-y'|^{D-2}}I + \dots \tag{88}$$

for a bulk scalar field; this is the most singular term.[15]

For the charged scalars framed in way described above, the Wilson lines used in the framing remain after the OPE contraction. Only at the bifurcation surface can one have a contraction of this type between purely left and purely right operators. Therefore, we can define

$$W_{LR}(\hat{x}_L,\hat{x}_R) = \lim_{y\to\mathscr{B},L}\lim_{y'\to\mathscr{B},R}|y-y'|^{D-2}\hat\phi_L^\dagger(y)\hat\phi_R(y')\,, \tag{89}$$

---

[15]For this singularity to be present, we should first take the limit $N\to\infty$, and then $y\to y'$.

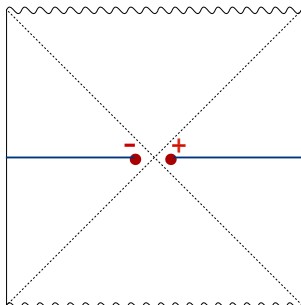

Figure 7: Construction of the Wilson line via OPE fusion at the bifurcation surface of a negatively charged operator from the left and a positively charged operator from the right.

where the limits are taken to the bifurcation surface, $\mathscr{B}$, from the left and the right. It is clear that this operator satisfies the correct commutation relations, since they are derived from bulk perturbation theory, and so must be consistent with the bulk OPE. Note that around general states (not close to the thermofield state), this limit has no divergence, and so no $W_{LR}$ can be extracted.

One may wonder whether this representation of the Wilson line acting on the thermofield double state agrees with the expression we found in the previous section. To see this, remember that the action of any left operator on the thermofield double state can be replaced by the action of a right operator whose insertion time is $t + i\beta/2$ [12]. Using this, the expression (89) can be rewritten as the coefficient of the divergence of two right bulk operators of opposite charges inserted at the *same* radial position close to the bifurcation surface, but at times 0 and $i\beta/2$. At the horizon, $g_{tt}$ vanishes, so the two points are very close together; thus, we can again use the bulk OPE to replace the two scalar insertions by the identity. As for the attached Wilson lines, if they were originally stretching along the $t = 0$ surface, now they will stretch from $r_+(1 + \epsilon)$ to infinity along the $t = 0$ line and from infinity to $r_+(1 + \epsilon)$ along $t = i\beta/2$. This yields exactly the same Wilson line that we were computing in the previous section.

## 4.3 Action on gauge-shifted states

In this subsection, we would like to make a connection to the work of [15] on mirror operators and state-dependence in the eternal black hole. The authors considered a set of time-shifted states

$$|\Psi_T\rangle = e^{iH_L T}|\Psi\rangle_{tfd} \tag{90}$$

whose gravity dual differs from the usual eternal black hole only by a large diffeomorphism.

Their argument pro state dependence consisted of two parts. First, by considering relational observables (which is just the statement that the bulk field should be gauge invariant), [15] showed that the mirror operator had to depend on the gauge parameter $T$. Then, they showed that by taking $T$ to be exponentially large, there didn't exist a state-independent operator that behaved correctly in all the time-shifted states.

In this section, we discuss the analogue of this argument for the case of electrically (rather than gravitationally) charged operators. As we will show, the dependence of the mirror operators on the large gauge parameter can be entirely understood in terms of the Wilson line, which also predicts certain corrections to the expression proposed in [15]. The advantage of thinking about the Wilson line is that the entire problem of state-dependence is shifted to a single object, whose presence - as we have argued - is already required by the bulk-to-boundary dictionary in the eternal black hole. However, we will not find any state-dependence in our

electromagnetic analogue. We understand this as a consequence of the gauge group being compact.

The analogue of the time-shifted states in our electromagntic setup are the "gauge-shifted" states

$$|\Psi_\lambda\rangle = e^{-iQ_L\lambda}|\Psi\rangle_{tfd} \,. \tag{91}$$

The states $|\Psi_\lambda\rangle$ can also be obtained via a path integral: one performs the same path integral over the Euclidean cylinder that produces the thermofield double state; however, when gluing the Euclidean geometry onto the Lorentzian CFTs, one has a choice of relative global charge rotation generated by $Q = \frac{1}{2}(Q_R - Q_L)$. While there is no natural "zero" of the charge rotation, the different states will have maximal entanglement between charged operators rotated by different phases. The zero mode of the Wilson line measures precisely this relative phase rotation. This construction is perfectly analogous with the path integral representation of the time-shifted states [15].

Using the path integral construction, the microscopic formula for $|\Psi_\lambda\rangle$ is

$$|\Psi_\lambda\rangle = \frac{1}{\sqrt{Z_\beta}} \sum_E e^{-\beta E/2} e^{iq\lambda} |E,-q\rangle_L |E,q\rangle_R \,, \tag{92}$$

where $q$ is the charge of the of the microstate of energy $E$ and we are assuming the energy spectrum is non-degenerate. The correlated structure of the charges is due to the fact that $(Q_L + Q_R)|\Psi_\lambda\rangle = 0$ [16].

The expressions in [15] are valid in the approximation in which all gravitational dressing of the scalar operator is neglected, except for the commutator with the Hamiltonian. In our language, this means that only the zero mode of the Wilson line is kept. As is evident from our discussion in section 2, this is not quite a consistent approximation (the non-zero modes have the same scaling with the gauge coupling), but it does capture the essential part of the physics, i.e. it has the correct commutators with the boundary charges. We can then translate the results of [15] into our Wilson line language, with the replacement

$$q \leftrightarrow \omega \,, \quad \lambda \leftrightarrow T \,, \tag{93}$$

where $\lambda$ is the zero mode of the unexponentiated Wilson line $\varphi$, and $q$ is the charge of the bulk field in question.

Using the fact that (in the gravity approximation), the set of states $|\Psi_T\rangle$ are almost orthogonal for different values of $T$, [15] wrote an expression for the mirror operators that has the expected behaviour within correlation functions. A simplified version of this expression in Fourier space is

$$\widetilde{\mathcal{O}}_\omega = \int_{-T_{cut}}^{T_{cut}} dT \, e^{i\omega T} \mathcal{O}_{\omega,L} P_{\Psi_T} \,, \tag{94}$$

where $P_{\Psi_T}$ is the projector onto the small Hilbert space $\mathscr{H}_{\Psi_T}$, satisfying $P_{\Psi_T} \mathcal{O}_\gamma |\Psi_T\rangle = \mathcal{O}_\gamma |\Psi_T\rangle$.

In our case, the overlap of the $\lambda$-shifted states can be estimated to be

$$\langle\Psi_\lambda|\Psi_{\lambda'}\rangle = \frac{1}{Z_\beta} \sum_E e^{-\beta E} e^{iq(\lambda'-\lambda)} \approx e^{-N(\lambda-\lambda')^2} \,, \tag{95}$$

where $N \propto k$ in three dimensions and $N \propto 1/e^2$ in higher $D$. This shows that the $\lambda$-shifted states are almost orthogonal for $|\lambda - \lambda'| > N^{-\frac{1}{2}}$.

Now, by analogy with the arguments of [15], we find that in the $\lambda$-shifted states, the mirror operators behave as $\mathcal{O}_L e^{iq\lambda}$. Thus, on the ensemble of states $|\Psi_\lambda\rangle$, the mirror operator is given

by (94) with the replacement (93), where $\lambda$ is integrated from 0 to $2\pi$. We would like to compare this expression with the mirror operators that we found:

$$\widetilde{\mathscr{O}} = \mathscr{O}_L \, W_{LR} \,. \tag{96}$$

As discussed, in order to compare with [15], we only need to consider the zero mode of $W_{LR}$ in the expression above. When acting on the thermofield double state, this zero mode is precisely given by the spatial average we considered in section 4.1. Using our expressions from section 4.1, the action of the Wilson line $W_{LR}$ on $\mathscr{H}_{\Psi_\lambda}$ is given by

$$W_{LR}|\Psi_\lambda\rangle = W_{LR}e^{-iQ_L\lambda}|\Psi\rangle_{\text{tfd}} = e^{iq\lambda}e^{-iQ_L\lambda}W_{LR}|\Psi\rangle_{\text{tfd}} = e^{iq\lambda+q\alpha Q_L}|\Psi_\lambda\rangle \,, \tag{97}$$

$$W_{LR}\mathscr{O}_R|\Psi_\lambda\rangle = [W_{LR}, \mathscr{O}_R]|\Psi_\lambda\rangle + \mathscr{O}_R e^{iq\lambda-\beta Q}|\Psi_\lambda\rangle = [W_{LR}, \mathscr{O}_R]|\Psi_\lambda\rangle + e^{iq\lambda+q\alpha Q_L}\mathscr{O}_R|\Psi_\lambda\rangle \,, \tag{98}$$

where $\alpha = 2\pi\beta/k$ in three dimensions and $\beta e^2/2r_+$ in higher $D$.

We see from the above expressions that, due to the non-trivial action of the zero mode of the Wilson line on the thermofield double state, the $\lambda$-dependence of the mirror operator is not just $e^{iq\lambda}$, but there is an additional shift $e^{\alpha q Q_L}$. In other words - the $\lambda$-shifted states are not eigenstates of the Wilson line operator, as (94) seems to indicate. Moreover, when the mirror operator acts on non-trivial elements of the small Hilbert space around $|\Psi_\lambda\rangle$, there is also a commutator term, which in general will not vanish. The action of the Wilson line on the $\lambda$-shifted states becomes even more complicated if we keep all the modes of the Wilson line, as we should in order to have a consistent approximation. Using the methods presented in sections 4.1, 2.3 and 3.3, this action can in principle be written down entirely explicitly.

Now, let us comment on the issue of state dependence. In [15], state dependence was due to the fact that, since $T$ could be taken to be arbitrarily large, there were many more states $|\Psi_T\rangle$ than the dimension of the Hilbert space, so one could derive a contradiction. In particular, the expression (94) breaks down for $T$ very large, for reasons nicely explained in [14, 15]. Another way to see state dependence was that by integrating over very long times, one would project onto energy eigenstates, which then lead to a contradiction because such states are not expected to have a smooth horizon.

In our case, the gauge parameter is compact, $\lambda \sim \lambda + 2\pi$, and thus an appropriately modified analogue of the expression (94) will work for all $\lambda$, at least as far as the exponentiated Wilson line is concerned.[16] Therefore, in order to see state-dependence in our setup, we should study instead how the Wilson line behaves in the time-shifted states, i.e. we should consider its gravitational dressing.

It is easiest to derive a contradiction for the unexponentiated spatially-averaged Wilson line[17], $\bar\varphi$. Taking the expectation value of the commutator (80) in the time-shifted states (90) and expanding in the energy eigenbasis, we find

$$2\pi i R = \langle\Psi_T|[Q, \bar\varphi]|\Psi_T\rangle = \frac{1}{Z_\beta}\sum_{E,E'}e^{-\frac{\beta}{2}(E+E')+i(E-E')T}(q'-q)\langle E', -q'|_L\langle E', q'|_R \; \bar\varphi \; |E, -q\rangle_L|E, q\rangle_R \,. \tag{99}$$

---

[16] One can also ask whether the unexponentiated Wilson line operator $\varphi$, which appears to be perfectly well-behaved around each of the states $|\Psi_\lambda\rangle$, continues to be well-defined on the ensemble of all such states The answer is clearly no: $\varphi$ is not a globally well-defined operator because it is compact; another way to say this is that it is not gauge-invariant under integer-valued relative gauge transformations between the two boundaries.

What is interesting to note is that the same kind of arguments that imply a contradiction in having a globally defined linear operator in the gravitational case here imply that $\varphi$ is not globally well-defined - there is a contradiction between the commutation relation $[Q, \varphi] = i$, which is supposed to be valid in each of the states $|\Psi_\lambda\rangle$, and the expectation value of this commutator in the zero-charge eigenstate $\oint d\lambda|\Psi_\lambda\rangle$. This suggests that subtleties in defining non-perturbatively diffeomorphism-invariant gravitational analogs of the Wilson line operators in the time-shifted states are important for their state dependence.

[17] To be more precise, we consider a periodic function of $\bar\varphi$ that is very close to $\bar\varphi$ between $-\pi$ and $\pi$, so that the operator is gauge invariant. In all of the states we are about to consider, the value of $\bar\varphi$ is close to 0, so the behavior of the periodic function close to $\pm\pi$ will not affect the argument.

This equality is meant to hold in the domain of validity of the bulk analysis, up to exponentially small corrections. Since the left-hand side is independent of $T$, while the right-hand side is a sum over terms with frequencies $E - E'$, this relationship cannot hold for $T$ arbitrarily large.

Note that the sum is dominated by microstates with energies of order the black hole energy, whose level spacing is of order $e^{-N}$. Therefore, if $\bar{\varphi}$ has the minimum possible width, $\delta_E \sim e^{-N}$, in the energy eigenbasis, then (4.31) can be valid for a range of $T$ of up to order $\delta_E^{-1}$. For time-shifted states beyond this window, one will find that a different operator (as specified by its matrix elements in the energy eigenbasis) obeys (4.31).

An alternative to this state-dependent construction of the Wilson line operators was described by [19], in which the bulk gauge field is emergent at a scale below the Planck scale. The Wilson lines in the exponentially time-shifted states will be curved in the interior, so that in that scenario, one will exit the domain of validity of the gauge field description.

# 5 Discussion

In this article, we have shown that in order to correctly reproduce bulk perturbation theory in presence of charged operators in the background of an eternal black hole, a new gauge-invariant operator needs to be included in the holographic dictionary, namely a boundary-to-boundary Wilson line.

This operator appears to only exist around entangled states of the two CFTs that are dual to connected two-sided geometries, which suggests it is a state-dependent operator. Due to the factorized structure of the microscopic Hilbert space, this operator can be written as a (sum of) products of a charged operator from the left CFT, and an oppositely charged operator from the right. However, which left/right operator pair represents the Wilson line seems to depend on the state of the system.

We have studied various properties of the Wilson line, such as its relation to the CFT currents and its operator algebra; in particular, we showed that it behaves as a local operator from the point of view of either boundary CFT. In the special case of a three-dimensional bulk, we showed that the (unexponentiated) Wilson line obeys the same operator algebra as a non-chiral boson, but its action on the thermofield double state is non-trivial.

Our work provides a systematic way to incorporate $1/N$ corrections into the expression for the bulk field in the eternal black hole background. In particular, it clarifies the relation between mirror operators in single-sided black hole backgrounds and left operators in the eternal black hole: as we explained in the previous section, the mirror operators used in the reconstruction of a bulk field framed to the right boundary behave (96) as local left operators connected to the right boundary via the Wilson line. Since the Wilson line does not in general commute with the right operators, this will lead to modifications to the defining properties of the mirror operators [13] already at the first order[18] in $1/N$. Our methods determine this commutator to any desired order in perturbation theory.

The expression (96) suggests that when taking into account general $1/N$ corrections, it may be natural to split the construction of mirror operators in the single-sided black hole into two steps. In the first step, one finds a (left) mirror operator that commutes, when acting on $\mathcal{H}_\Psi$, with all the right operators, including the conserved charges; these are the analogues of the $\mathcal{O}_L$ in the eternal black hole to any order in $1/N$. In particular, one constructs mirror conserved charges and a mirror Hamiltonian. In the second step, one defines an operator that behaves as the Wilson line. The advantages of this two-step procedure would be that the first step

---

[18]Strictly speaking, the non-trivial commutators of the mirror operators with the boundary Hamiltonian postulated in [13] already represent such a $1/N$ correction.

simply amounts to applying a Tomita-Takesaki type construction to the algebra[19] generated by the right operators to arbitrary order in $1/N$, and that all the non-trivial commutators of the (right-framed) mirror operators with the right CFT operators are encoded in a single object: the Wilson line. It would be interesting to understand whether such a two-step construction emerges naturally from a Tomita-Takesaki type construction applied to systems with a global symmetry.

It would be very interesting to extend these results to gravity. There, the framing of gauge-invariant operators is much more complicated than in gauge theory - see [26] for a recent discussion. However, for the particular case of three dimensions, the bulk theory reduces at low energies to pure Einstein gravity in AdS$_3$, which can be rewritten as two copies of $SL(2,\mathbb{R})$ Chern-Simons theory [27, 28]. The bulk low energy sector that is described by the weakly coupled Chern-Simons is dual, in the eternal black hole background, to two copies of non-chiral Liouville theory. Each copy is associated to one of the $SL(2,\mathbb{R})$ factors and involves currents from *both* boundaries, as well as left to right Wilson lines. In this case, by integrating over a large range of gauge shifted states (which can now be interpreted as time shifted), one can asymptotically project onto energy eigenstates. Unlike in the gauge charge situation, such eigenstates must be essentially factorized, implying that no single operator can approximate this Liouville operator algebra on all of the shifted states.

In other words, although the boundary-to-boundary Wilson lines would seem to be well described by the weakly coupled Chern-Simons approximation in all the time-shifted states, this approximation must eventually break down in any complete theory of gravity in AdS$_3$.[20] As discussed at the end of section 4.3, it is possible that subtleties in defining these Wilson lines in a way that is both diffeomorphism invariant and remains in the weakly coupled Chern-Simons approximation (for example, so that no sharp Planckian features appear in the path of the Wilson line even for very long time shifts) for all of the time shifted states is important in this breakdown. It would be very interesting to further explore this issue.

### Acknowledgements

The authors are grateful to Sheer El-Showk, Benoit Estienne, Daniel Harlow, Suvrat Raju, Herman Verlinde and Konstantin Zarembo for interesting discussions. M.G. is grateful to the Center of Mathematical Sciences and Applications at Harvard University, where this work was initiated, for its support and hospitality. Her work is supported in part by the ERC advanced grant No 341222. The work of D.L.J. is supported in part by NSFCAREER grant PHY-1352084.

## A  Dirac quantization of $U(1)$ Chern-Simons

We work out the Dirac bracket quantization of the CS gauge field in the gauge $\partial_{\bar{z}} A_{\bar{z}} = 0$, adapted to the presence of two boundaries, and check that it automatically produces a Wilson line operator that is charged. This shows that as long as we correctly pick the gauge, bulk perturbation theory will produce the correct charges for the bulk fields.

We first consider the case of pure Chern-Simons theory, and then we couple it to a charged scalar field. The full action is given by (7).

---

[19]Modulo caveats [13] due to the fact that the right operators do not exactly form an algebra.

[20]Of course, near any such state, the Liouville approximation gives the correct operator algebra; it is just that these are not realized as globally well-defined linear operators.

## A.1 Pure Chern-Simons

In the coordinates (30), the variation of the action reads[21]

$$\delta S_{on-shell} = \frac{k}{8\pi} \int_{z=0} dx^+ dx^- (A_+ \delta A_- - A_- \delta A_+) - \frac{k}{8\pi} \int_{z=a} dx^+ dx^- (A_+ \delta A_- - A_- \delta A_+) \,. \quad (100)$$

We would like to fix $A_- = 0$ at both boundaries, which can be achieved by adding the boundary terms

$$S_{bnd} = \frac{k}{8\pi} \int_{z=0} dx^+ dx^- A_+ A_- - \frac{k}{8\pi} \int_{z=a} dx^+ dx^- A_+ A_- \,. \quad (101)$$

After adding these, the action can be brought to the simple form

$$S + S_b = \frac{k}{4\pi} \int d^3 x \left[ A_z \partial_+ A_- + A_+ (\partial_- A_z - \partial_z A_-) \right] \,. \quad (102)$$

One can then proceed to quantizing this action, e.g. à la Dirac. The momenta conjugate to $A_M$ are constrained:

$$\pi^+ = \pi^- = 0 \,, \qquad \pi^z - \frac{k}{4\pi} A_+ = 0 \,, \quad (103)$$

and the Gauss law, which is a secondary constraint, simply reads

$$\chi_1 = F_{+z} = 0 \quad (104)$$

Two of these constraints are first class: $\pi^-$ and the combination

$$\Omega = F_{+z} - \frac{4\pi}{k} \left( \partial_+ \pi^+ + \partial_z (\pi^z - \frac{k}{4\pi} A_+) \right) \,, \quad (105)$$

while the rest are second class. It is useful to perform the Dirac procedure in two steps, by first eliminating the conjugate variables $\pi^-, A_-$ (which decouple from the rest) and the momenta $\pi^+, \pi^z$, and only then gauge fixing. After the first step, the only non-trivial commutator is

$$\{A_+(x^+, z), A_z(x'^+, z')\} = -\frac{4\pi}{k} \delta(x^+ - x'^+) \delta(z - z') \,, \quad (106)$$

but we are still left with the Gauss law constraint (104), which is first class. To make it second class, we will be imposing the gauge condition

$$\chi_2 = \partial_z A_z = 0 \,, \quad (107)$$

which, as we argued in the main text, is compatible with the boundary conditions we want to impose. The Poisson bracket of the constraints is

$$\{\chi_1, \chi_2\} \equiv C_{12} = \frac{4\pi}{k} \partial_z \partial_{z'} \delta(z - z') \delta(x^+ - x'^+) \,. \quad (108)$$

The Dirac brackets are constructed as

$$\{f, g\}_{D.B.} = \{f, g\} - \int \{f, \chi_i\} (C^{-1})^{ij} \{\chi_j, g\} \,, \quad (109)$$

where $C^{-1}$ is the inverse of the constraints matrix. Denoting

$$(C^{-1})^{ij} = \frac{k}{4\pi} K(z, z') \epsilon^{ij} \delta(x^+ - x'^+) \,, \quad (110)$$

---

[21]We use conventions $\epsilon^{+-z} = 1$.

with $\epsilon^{12} = 1$, we find that

$$\partial_z^2 K(z,z') = \partial_{z'}^2 K(z,z') = \delta(z - z') \,, \tag{111}$$

with solution

$$K(z,z') = (z - z')\Theta(z - z') + \alpha(z')z + \beta(z') \,, \tag{112}$$

where $\alpha(z'), \beta(z')$ are linear functions of $z'$. This kernel acts nicely on any function that does not have poles in $z - z'$. In fact, the requirement that $C^{-1}C\lambda = \lambda$ for any doublet of functions $\lambda^T = \begin{pmatrix} \lambda_1 & \lambda_2 \end{pmatrix}$ completely fixes $\alpha(z')$ and $\beta(z')$, since

$$\int dz' dz'' K_{12}(z,z') C_{21}(z',z'') \lambda_1(z'') = \lambda_1(z) + (\alpha(a)z + \beta(a))\lambda_1'(a) - (z(\alpha(0) + 1) + \beta(0))\lambda_1'(0) \,. \tag{113}$$

Requiring that the terms proportional to $\lambda'$ vanish fixes

$$\alpha(a) = \beta(a) = \alpha(0) + 1 = \beta(0) = 0 \,, \tag{114}$$

which determines the linear functions $\alpha(z'), \beta(z')$. The final expression for $K(z,z')$ is

$$K(z,z') = (z - z')\Theta(z - z') + \left(\frac{z'}{a} - 1\right)z \,. \tag{115}$$

The Dirac bracket of $A_+$ with itself is

$$\begin{aligned}
\{A_+(x^+,z), A_+(x'^+,z')\}_{D.B.} &= \frac{4\pi}{k}[\partial_{z'}K(z,z') + \partial_z K(z,z')]\partial_{x^+}\delta(x^+ - x'^+) \\
&= \frac{4\pi}{k}\left(\frac{z + z'}{a} - 1\right)\partial_{x^+}\delta(x^+ - x'^+) \,. \tag{116}
\end{aligned}$$

On the other hand, the bulk-boundary dictionary (33)

$$A_+(x^+,z) = \frac{2}{k}\left[j_+^L(x^+) + \frac{z}{a}\left(j_+^R(x^+) - j_+^L(x^+)\right)\right] \tag{117}$$

yields

$$\begin{aligned}
[A_+(x^+,z), A_+(x'^+,z')] = &\frac{4}{k^2}[j_+^L(x^+), j_+^L(x'^+)]\left(1 - \frac{z + z'}{a}\right) + \\
&\frac{4}{k^2}\frac{zz'}{a^2}\left([j_+^L(x^+), j_+^L(x'^+)] + [j_+^R(x^+), j_+^R(x'^+)]\right) \,. \tag{118}
\end{aligned}$$

This expression matches (116) provided that

$$[j_+^L(x^+), j_+^L(x'^+)] = -[j_+^R(x^+), j_+^R(x'^+)] = -i\pi k\, \partial_{x^+}\delta(x^+ - x'^+) \,, \tag{119}$$

which can be checked agrees with the usual current-current OPE. The difference in signs between the $j_L$ and $j_R$ commutators is due to the different choice of orientation of the right boundary.

The other non-zero Dirac bracket is

$$\{A_+(x^+,z), A_z(x'^+,z')\}_{D.B.} = -\frac{4\pi}{k}\delta(x^+ - x'^+)[\delta(z - z') + \partial_z\partial_{z'}K(z,z')] = -\frac{4\pi}{ka}\delta(x^+ - x'^+) \,, \tag{120}$$

from which we can find the commutator of the nonchiral boson $\varphi = aA_z$ defined in (34) with the CFT currents

$$[j_+^L(x^+), \varphi(x'^+)] = [j_+^R(x^+), \varphi(x'^+)] = -2\pi i\, \delta(x^+ - x'^+) \,, \tag{121}$$

which is perfectly consistent with (35) and the current-current commutator. The commutators of the conserved charges $Q_L = \frac{1}{2\pi}\int j_+^L(x^+)dx^+$, $Q_R = -\frac{1}{2\pi}\int j_+^R(x^+)dx^+$ with $\varphi$ are thus

$$[Q_R, \varphi] = -[Q_L, \varphi] = i \,. \tag{122}$$

## A.2  Coupling to matter

Let us now couple the Chern-Simons theory to a matter current $J^\mu$. To be specific, we will take the matter to be a complex scalar field, with the total action given by (7). We assume that the only non-zero components of the metric are $g_{+-}$ and $g_{zz}$. There are now two new primary constraints [29], in addition to (103)

$$\pi_\phi + (D^-\phi)^\star \sqrt{g} = 0\,, \qquad \pi_{\phi^\star} + D^-\phi \sqrt{g} = 0\,, \tag{123}$$

and the Gauss law constraint now reads

$$\chi_1' = F_{+z} + \frac{4\pi}{k}\sqrt{g}\, J^- = F_{+z} + \frac{4\pi iq}{k}(\phi\,\pi_\phi - \phi^\star \pi_{\phi^\star})\,. \tag{124}$$

The first class constraints are $\pi^-$ and the combination

$$\Omega' = F_{+z} + \frac{4\pi iq}{k}(\phi\,\pi_\phi - \phi^\star \pi_{\phi^\star}) - \frac{4\pi}{k}\left(\partial_+\pi^+ + \partial_z(\pi^z - \frac{k}{4\pi}A_+)\right)\,. \tag{125}$$

The non-trivial equal-time Poisson brackets are

$$\{\pi^+, \pi^z - \frac{k}{4\pi}A_+\}_{P.B.} = \frac{k}{4\pi}\delta(x^+ - x'^+)\delta(z - z')\,,$$
$$\{\pi^+, \pi_\phi + (D^-\phi)^\star \sqrt{g}\}_{P.B.} = -iqg^{+-}\sqrt{g}\,\phi^\star\,\delta(x^+ - x'^+)\delta(z - z')\,,$$
$$\{\pi^+, \pi_{\phi^\star} + D^-\phi \sqrt{g}\}_{P.B.} = iqg^{+-}\sqrt{g}\,\phi\,\delta(x^+ - x'^+)\delta(z - z')\,,$$
$$\{\pi_\phi + (D^-\phi)^\star \sqrt{g}, \pi_{\phi^\star} + D^-\phi \sqrt{g}\}_{P.B.} = (\partial_+ - \partial_+')\delta(x^+ - x'^+)\delta(z - z')g^{+-}\sqrt{g} +$$
$$2iqA_+ g^{+-}\sqrt{g}\delta(x^+ - x'^+)\delta(z - z')\,,$$
$$\{\pi_{\phi^\star} + D^-\phi \sqrt{g}, \pi_\phi + (D^-\phi)^\star \sqrt{g}\}_{P.B.} = (\partial_+ - \partial_+')\delta(x^+ - x'^+)\delta(z - z')g^{+-}\sqrt{g} -$$
$$2iqA_+ g^{+-}\sqrt{g}\delta(x^+ - x'^+)\delta(z - z')\,. \tag{126}$$

Imposing the gauge-fixing condition $\gamma = \partial_z A_z = 0$, we find two additional brackets

$$\{\partial_z A_z, \pi^z - \frac{k}{4\pi}A_+\}_{P.B.} = \delta(x^+ - x'^+)\partial_z\delta(z - z')\,,$$
$$\{\partial_z A_z, \Omega'\}_{P.B.} = -\frac{4\pi}{k}\delta(x^+ - x'^+)\partial_z\partial_{z'}\delta(z - z')\,. \tag{127}$$

We are interested in the Dirac brackets of the Wilson line $\varphi = a A_z$, which are given by

$$\{A_z(y), A_+(y')\}_{D.B.} = (C^{-1})^{z+}(y, y') + \frac{4\pi}{k}\partial_z(C^{-1})^{\Omega+}(y, y')\,,$$
$$\{A_z(y), \phi(y')\}_{D.B.} = (C^{-1})^{z\phi}(y, y') + \frac{4\pi}{k}\partial_z(C^{-1})^{\Omega\phi}(y, y')\,, \tag{128}$$

where $(C^{-1})^{ij}$ denote the respective components of the inverse matrix of constraints and $y, y'$ label bulk points with $x^- = x'^-$. We have the following relations among its components

$$(C^{-1})^{z+}(y, y') = \frac{4\pi}{k}\delta(y - y')\,, \qquad (C^{-1})^{\Omega+}(y, y') = \frac{4\pi}{k}\partial_{z'}(C^{-1})^{\Omega\gamma}(y, y')\,, \tag{129}$$

where $(C^{-1})^{\Omega\gamma}(y, y')$ satisfies

$$\partial_z^2(C^{-1})^{\Omega\gamma}(y, y') = \partial_{z'}^2(C^{-1})^{\Omega\gamma}(y, y') = \frac{k}{4\pi}\delta(y - y')\,. \tag{130}$$

A careful analysis along the lines of the previous section yields

$$(C^{-1})^{\Omega\gamma}(y, y') = \frac{k}{4\pi}K(z, z')\delta(x^+ - x'^+)\,, \tag{131}$$

where $K(z, z')$ is given in (115). Thus, we find that the (equal-time) Dirac bracket of $A_z$ with $A_+$ is given by exactly the same expression (120) as in the previous section. We also have

$$\partial_{+'}(C^{-1})^{\Omega\phi}(y, y') - iqA_+(y')(C^{-1})^{\Omega\phi}(y, y') = \frac{iq}{2}\phi(y')(C^{-1})^{\Omega+}(y, y'),$$

$$\partial_{+'}(C^{-1})^{z\phi}(y, y') - iqA_+(y')(C^{-1})^{z\phi}(y, y') = \frac{2\pi iq}{k}\phi(y')\delta(y - y').\tag{132}$$

Using these relations, we find that the commutator of the Wilson line with the scalar field satisfies

$$\partial_{+'}\{\varphi(y), \phi(y')\}_{D.B.} - iqA_+(y')\{\varphi(y), \phi(y')\}_{D.B.} = \frac{2\pi iqa}{k}\phi(y')\left[\delta(y - y') + \partial_z(C^{-1})^{\Omega+}(y, y')\right],$$

$$= \frac{2\pi iq}{k}\phi(y')\delta(x^+ - x'^+).\tag{133}$$

Solving this equation to lowest order, we find that

$$\{\varphi(y), \phi(y')\}_{D.B.} = -\frac{2\pi iq}{k}\phi(y')\Theta(x^+ - x'^+),\tag{134}$$

which is perfectly consistent with the commutators of the scalar field and the currents.

Let us now consider the commutator

$$\{\phi(y), A_+(y')\}_{D.B.} = \frac{4\pi iq}{k}(C^{-1})^{\Omega+}(y, y')\phi(y) + \frac{4\pi}{k}\partial_{+'}(C^{-1})^{\phi\Omega}(y, y').\tag{135}$$

The last term is a total derivative and it will not contribute to the commutator with the boundary charges, so let us just drop it for now. We obtain

$$\{\phi(y), A_+(y')\}_{D.B.} = \frac{4\pi iq}{k}\phi(y)\partial_{z'}K_{12}(z, z')\delta(x^+ - x'^+)$$

$$= \frac{4\pi iq}{k}\phi(y)\left(\frac{z}{a} - \Theta(z - z')\right)\delta(x^+ - x'^+).\tag{136}$$

The gauge-invariant operators connected by a Wilson line to either boundary are

$$\hat{\phi}_L = e^{iq\int_z^0 A_z(x, z')dz'}\phi(x, z) = e^{-iqzA_z(x)}\phi(x, z),\tag{137}$$

$$\hat{\phi}_R = e^{iq\int_z^a A_z(x, z')dz'}\phi(x, z) = e^{iq(a-z)A_z(x)}\phi(x, z).\tag{138}$$

where we have used the fact that $A_z = const$. Their commutators with the bulk gauge field are

$$[\hat{\phi}_L(y), A_+(y')] = -\frac{4\pi iq}{k}\hat{\phi}_L(x^+, x^-, z)\Theta(z - z')\delta(x^+ - x'^+),\tag{139}$$

$$[\hat{\phi}_R(y), A_+(y')] = \frac{4\pi iq}{k}\hat{\phi}_R(x^+, x^-, z)(1 - \Theta(z - z'))\delta(x^+ - x'^+).\tag{140}$$

Setting $z' = 0$ or $z' = a$ we can find the commutator with the currents on the two boundaries, $j_L(x^+) = \frac{k}{2}A_+(x^+, 0)$ and $j_R(x^+) = \frac{k}{2}A_+(x^+, a)$, which are as expected

$$[\hat{\phi}_L(y), j_L(x'^+)] = -2\pi iq\hat{\phi}_L(y)\delta(x^+ - x'^+), \qquad [\hat{\phi}_L(y), j_R(x'^+)] = 0.\tag{141}$$

## B  Global coordinates in three dimensions

In this appendix, we explicitly perform the change of coordinates between the Schwarzschild coordinates (50) to the global coordinates (52) using the boundary-to-boundary geodesics depicted in figure 5 in the simplest case of three bulk dimensions.

The BTZ black hole metric reads

$$ds^2 = -\frac{r^2 - r_+^2}{\ell^2} dt^2 + \frac{\ell^2 dr^2}{r^2 - r_+^2} + r^2 dx^2 \,. \tag{142}$$

We concentrate on a set of geodesics at constant x, which satisfy

$$\dot{t}(r^2 - r_+^2) = -E \, r_+ \ell^2 \,, \quad \dot{r}^2 = r^2 - r_+^2 + E^2 r_+^2 \tag{143}$$

for some dimensionless constant $E$. The solution is

$$r(\lambda) = r_+ \cosh \lambda - \frac{r_+ E^2}{2} e^{-\lambda} \,, \quad t(\lambda) = t_0 + \frac{\ell^2}{2r_+} \ln \frac{e^{2\lambda} - (1-E)^2}{e^{2\lambda} - (1+E)^2} \,. \tag{144}$$

The geodesic will penetrate the horizon if $0 < |E| < 1$. The minimum of the radial coordinate $r$ on this geodesic is $r_{\min} = r_+ \sqrt{1 - E^2}$, which occurs at $\lambda = \frac{1}{2} \ln(1 - E^2)$. Requiring that $r_{\min}$ is reached on the symmetry line (inside the horizon) at $t = 0$ fixes[22]

$$t_0 = \frac{\ell^2}{2r_+} \ln \frac{1+E}{1-E} \,. \tag{145}$$

which implies that $\lim_{\lambda \to \pm\infty} t(\lambda) = \pm t_0$, and thus the geodesic extends symmetrically between the two boundaries. It is thus useful to introduce a new affine coordinate

$$\sigma = \lambda - \frac{1}{2} \ln(1 - E^2) \,, \tag{146}$$

which will have its zero on the symmetry line. The metric can now be rewritten in terms of $\sigma$ and the parameters $E$ or $t_0$

$$\frac{ds^2}{\ell^2} = d\sigma^2 - \frac{dE^2}{(1-E^2)^2} \frac{r^2}{r_+^2} + \frac{r^2}{\ell^2} dx^2 \,, \tag{147}$$

where $r = r(\sigma, E)$, or

$$\frac{ds^2}{\ell^2} = d\sigma^2 - \frac{r^2}{r_+^2} d\tau^2 + \frac{r^2}{\ell^2} dx^2 \,, \tag{148}$$

where $\tau = t_0 \, r_+ / \ell^2$ and

$$r(\sigma, \tau) = r_+ \frac{\cosh \sigma}{\cosh \tau} \,. \tag{149}$$

The $\tau = const.$ surfaces are hyperboloids. The minimum value of $r$ on a constant $\tau$ hypersurface is $r_{min} = r_+ / \cosh \tau$ and it occurs at $\sigma = 0$. The constant $\tau$ hypersurface crosses the horizon at $\sigma = \pm \tau$. To obtain the full change of coordinates $(t, r) \to (\tau, \sigma)$, note that $dt$ is given by

$$dt^2 = \frac{\ell^4}{r^2 - r_+^2} \left( \frac{dr^2}{r^2 - r_+^2} + \frac{r^2 d\tau^2}{r_+^2} - d\sigma^2 \right) = \left( \frac{\ell^2 (d\tau \sinh 2\sigma - d\sigma \sinh 2\tau)}{2r_+ (\cosh^2 \sigma - \cosh^2 \tau)} \right)^2 \,. \tag{150}$$

Integrating, we find

$$t = \frac{\ell^2}{2r_+} \ln \frac{\sinh(\sigma + \tau)}{\sinh(\sigma - \tau)} \,. \tag{151}$$

Note that near the boundaries, we have $t = \pm \frac{\ell^2}{r_+} \tau$, as we should.

---

[22]It may be useful to remember that $\operatorname{arctanh} x = \frac{1}{2} \ln \frac{1+x}{1-x}$.

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
