# Peer review of "On the construction of charged operators inside an eternal black hole"

_SciPost Physics, doi:SciPost Phys. 3, 016 (2017)_

## Round 1 · Referee Report · Anonymous · 2017-1-23

Strengths

- Discusses an important question in the reconstruction of the bulk in AdS/CFT.

- Makes an important point about the necessity of including the boundary current to all orders in order to correctly reproduce the bulk algebra of operators at leading order in 1/k.

- An elegant discussion of charged operator reconstruction for bulk Chern-Simons theory, which enables a much more explicit treatment than is standard in the literature for more general gauge fields.

- Nicely complements an earlier paper of Harlow, which emphasized different aspects of the same problem.

Weaknesses

- The discussion of ``state-dependence'' is somewhat misleading, as explained in the report below.

- A few other misleading or incorrect statements, which are pointed out below.

Report

This is an excellent paper, which I recommend be published after the authors consider the following three points and make modifications as they feel are appropriate:

1) The authors use the term ``state dependence'' to mean something considerably milder than the use of this term by Papadodimas and Raju in work which they cite. This distinction has been previously emphasized in 1405.1995 and 1506.01337: the weaker version, which in 1506.01337 is called ``background dependence'', is a standard thing in quantum mechanics: we can have observables which make sense only on a linear subspace of the states in the Hilbert space. An example of this is quasi-particle excitations of a solid, which don't make sense in states where the solid has melted or evaporated. Similarly in the quantum error correction description of bulk emergence, we only expect the bulk operators to make sense in some ``code subspace''. In the paper under review, the two-sided Wilson lines only make sense in a subspace of states where the geometry is connected, so they are in the same class. This is to be contrasted with the more extreme situation advocated by PR where, contrary to quantum mechanics, the measurement theory of an observable is nonlinear. This is NOT required by anything discussed in this paper, and as such calling it state-dependence and citing PR is misleading.

2) On page 5 the authors suggest that the two-sided Wilson line behaves like a product of local operators, one in each CFT, but actually this is only true in the Chern-Simons case, since at leading order in $1/k$ we have $F=0$ throughout the bulk. It will not be true at higher order in $1/k$, or at all in Maxwell theory. One easy way to see this is that when $F$ is nontrivial, we can consider a Wilson line that for a while hugs the boundary near one of its endpoints. We can then easily send a signal to a point on the line from a boundary point which is spacelike-separated from both endpoints, and thus find an operator that does not commute with the line despite being spacelike-separated from its endpoints. Another way of saying this is that we can just reconstruct the electric field at some point on the line in a boundary region which is spacelike separated from the endpoints of the line, so again they can't commute and thus the line can't be localized at its endpoints. The authors need to clarify that the locality they mention is not what we usually expect, and they also need to modify section 3.3 which thus must be wrong.

3) In section 4.2, they suggest that the ``OPE'' construction of the boundary-boundary Wilson line, also advocated in 1510.07911, can only be accomplished when there is a bifurcate horizon. But as explained in 1510.07911, this is true only if we expect that bulk reconstruction is only possible in the causal wedge. In fact it is expected to work in the larger ``entanglement wedge'', see 1601.05416 and the references therein, in which case a bifurcate horizon is not required.

Requested changes

Points 1-3) given above need to be addressed, I leave it to the authors to decide how.

  • validity: high
  • significance: high
  • originality: good
  • clarity: high
  • formatting: excellent
  • grammar: perfect

Author:  Monica Guica  on 2017-05-05  [id 126]

(in reply to Report 1 on 2017-01-23)
Category:
reply to objection

We thank the referee for his comments. We have submitted a new version of our article, with the following modifications:

  • we specified in the introduction and at the beginning of section 3.3 that the Wilson line behaves as a local operator for the particular choice of bulk curve $\Gamma$ we discuss in that section. This should address comment 2) of the referee.

  • we added at the end of section 4.3 a brief proof of state-dependence (in the strong, Papadodimas-Raju sense) of the Wilson line in the time-shifted states and commented on the relationship with the proposal of 1510.07911. This should address comment 1).

  • we also specified at the beginning of section 4.2 that we are interested in a representation of the Wilson line in terms of simple operators in the two CFTs, for which only the case of a bifurcate horizon is relevant, thus addressing comment 3).

Attachment:

chopbh9.pdf

---

## Round 2 · Author Response

We thank the referee for his comments. We have submitted a new version of our article, with the following modifications:

  • we specified in the introduction and at the beginning of section 3.3 that the Wilson line behaves as a local operator for the particular choice of bulk curve Γ

we discuss in that section. This should address comment 2) of the referee.

  • we added at the end of section 4.3 a brief proof of state-dependence (in the strong, Papadodimas-Raju sense) of the Wilson line in the time-shifted states and commented on the relationship with the proposal of 1510.07911. This should address comment 1).

  • we also specified at the beginning of section 4.2 that we are interested in a representation of the Wilson line in terms of simple operators in the two CFTs, for which only the case of a bifurcate horizon is relevant, thus addressing comment 3).

---

## Round 2 · List of Changes

See comments above.

You are currently on this page

Resubmission 1511.05627v2 on 27 July 2017

---

## Editorial Decision

published